# Timing and pacing of the Late Devonian mass extinction event regulated by eccentricity and obliquity

David De Vleeschouwer [1,2], Anne-Christine Da Silva [3,4], Matthias Sinnesael[2], Daizhao Chen[5], James E. Day[6], Michael T. Whalen[7], Zenghui Guo[5] & Philippe Claeys [2]

The Late Devonian envelops one of Earth's big five mass extinction events at the Frasnian–Famennian boundary (374 Ma). Environmental change across the extinction severely affected Devonian reef-builders, besides many other forms of marine life. Yet, cause-and-effect chains leading to the extinction remain poorly constrained as Late Devonian stratigraphy is poorly resolved, compared to younger cataclysmic intervals. In this study we present a global orbitally calibrated chronology across this momentous interval, applying cyclostratigraphic techniques. Our timescale stipulates that 600 kyr separate the lower and upper Kellwasser positive $\delta^{13}$C excursions. The latter excursion is paced by obliquity and is therein similar to Mesozoic intervals of environmental upheaval, like the Cretaceous Ocean-Anoxic-Event-2 (OAE-2). This obliquity signature implies coincidence with a minimum of the 2.4 Myr eccentricity cycle, during which obliquity prevails over precession, and highlights the decisive role of astronomically forced "Milankovitch" climate change in timing and pacing the Late Devonian mass extinction.

[1] MARUM—Center for Marine Environmental Sciences, University of Bremen, Leobenerstraße, 28359 Bremen, Germany. [2] Analytical, Environmental and Geo-Chemistry (AMGC), Vrije Universiteit Brussel, Pleinlaan 2, 1050 Brussels, Belgium. [3] Sedimentary Petrology Laboratory, Liège University, Sart Tilman B20, Allée du Six Août 12, 4000 Liège, Belgium. [4] Paleomagnetic Laboratory, Utrecht University, Budapestlaan 17, 3584 CD Utrecht, The Netherlands. [5] Institute of Geology and Geophysics, Chinese Academy of Science, 19 Beitucheng Xilu, Chaoyang District, Beijing, 100029, China. [6] Department of Geography–Geology, Illinois State University, Normal, Illinois 61790-4400, USA. [7] Department of Geosciences, University of Alaska, Fairbanks, Alaska 99775-5780, USA. Correspondence and requests for materials should be addressed to D.D.V. (email: ddevleeschouwer@marum.de)

Frasnian–Famennian (F–F) boundary sections around the world are often characterized by two horizons of dark bituminous shale, the so-called lower and upper Kellwasser horizons (LKW and UKW)[1,2]. Both layers coincide with a positive δ[13]C excursion, as recognized in North America, Morocco, Europe, Russia, western Australia, and southern China[1,3–10]. The worldwide incidence of Kellwasser horizons indicates a major perturbation of the global carbon cycle, and is associated with one of the most prominent mass extinction events in Earth history, primarily affecting tropical marine benthos[11]. Stromatoporoid and coral reefs were ubiquitous during the Middle to Late Devonian, but never truly recovered after the Late Devonian mass extinction. Despite this major reshuffling of the Earth's system, the timing and pacing of the environmental changes responsible for the mass extinction remain poorly constrained. Recently, however, cyclostratigraphic efforts yielded constraints on the numerical ages of stage boundaries within the Devonian System[12–15]. These studies provided a new chronometer for this part of the Paleozoic, based on the most stable astronomical cycle: the 405-kyr long eccentricity cycle. In this study, we take the application of Devonian astrochronology to sub-eccentricity resolution. Therefore, we considered the magnetic susceptibility and carbon isotope series of six globally distributed F–F sections (Fig. 1). Two of these sections, Kowala (Poland) and Section C (western Canada), were the subject of previous cyclostratigraphic studies[15,16], yielding orbital chronologies based on 405-kyr long eccentricity cycles. We correlated the other four sections (from Belgium, China and the United States) into this astrochronologic framework, based on biostratigraphy, δ[13]C chemostratigraphy, and magnetic susceptibility proxy records. We then used a Monte-Carlo approach to further refine the original eccentricity-based astrochronology ("Methods"). The result is a common chronology for all six sections, i.e., a global astrochronology for the Late Devonian. This global chronology provides new insight into the rhythm of environmental change across the F–F transition, and reveals the obliquity signature of the UKW positive δ[13]C excursion. These results confirm the importance of astronomically forced climate change in pacing the biogeochemical interactions that led to the extinction event.

## Results

**Six globally distributed Frasnian–Famennian sections.** The six globally distributed F–F sections considered in this study are located in Poland (Kowala section), western Canada (Section C), Iowa (H-32 and CG-1 drill core), Belgium (Sinsin section), and South China (Fuhe section). The H-32, CG-1, and Sinsin proxy records were constructed in the framework of this study, whereas data from Section C[17,18], Kowala[16,19], and Fuhe[3,20,21] have been previously published (for details see Fig. 1 caption). All six sections were deposited in a tropical setting, albeit in different parts of the Panthalassic, Rheic and Paleo-Tethys Oceans (Fig. 1). We selected sections that were deposited in basin-to-slope deep-water environments, so to maximize stratigraphic continuity. The CG-1 drill core is the exception to this rule, as it constitutes the shelf-margin time-equivalent section of deep-water section H-32.

The magnetic susceptibility series of Section C (western Canada) served as the basis for the Frasnian astrochronology published in a previous study by De Vleeschouwer et al.[15]. In that paper, 16 Frasnian long eccentricity cycles (Fr-LECs, each 405 kyr in duration) were counted. This chronology was later refined by Whalen et al.[17], who revealed that half a 405-kyr long eccentricity cycle needed to be added to the chronology of Section C, after comparison with the contemporaneous Kowala section and by constructing a detailed δ[13]C record for the F–F interval (shown in Fig. 1). This revision reconciled the astrochronologic interpretations from Section C[15] and the Kowala[16], with a total of 16.5 Fr-LECs and with the LKW and UKW occurring in the 15th and 17th Fr-LEC, respectively. In this paper, we tie four additional sections (H-32, CG-1, Sinsin, and Fuhe) into the common astrochronologic framework of Section C and Kowala, so to attain a global astrochronology for the F–F mass extinction interval (Figs. 2–4). We establish the temporal relationships between the different sections by correlating distinct features in magnetic susceptibility (red ties in Fig. 5) and carbon isotope geochemistry (blue ties in Fig. 6), while respecting biostratigraphic constraints. The determination of stratigraphic relationships between globally distributed sections based on carbon isotope variations is well established[22], yet interbasinal correlations based on magnetic susceptibility data are not undisputed. This is because the magnetic susceptibility of a rock sample depends on the concentration and type of magnetic minerals (often assumed to be a function of sea-level and climate), as well as the size and shape of the magnetic grains. Nevertheless, several Devonian examples exist of valid global correlations based on magnetic susceptibility, despite differences in palaeogeographic setting, facies, accumulation rate or depositional history[3,23–25]. These examples suggest the importance of a magnetic susceptibility forcing mechanism operating at the global scale. For example, Section C, H-32, Sinsin, and Fuhe exhibit a characteristic double minimum in magnetic susceptibility just prior to the F–F boundary (Fig. 1). This susceptibility signature corroborates the utility of magnetic susceptibility as a correlative tool in the framework of this Late Devonian study.

As mentioned earlier, the magnetic-susceptibility and carbon-isotope tie-points (Figs. 5 and 6) were drawn respecting biostratigraphic constraints. However, in two instances, correlating lines traverse different biostratigraphic zones. Specifically, we correlate the stratigraphic levels at 19 and 21.5 m at the CG-1 core with 320 and 330 m at Section C (Fig. 5). These levels occur in Frasnian Conodont Zone 11 in the CG-1 core, but occur in Frasnian Conodont Zone 12 in Section C. In this case, it is unclear whether this discrepancy is caused by diachronism between Iowa and western Canada, an inaccurate biostatigraphic zonation, or erroneous correlation. Yet, the excellent match between the susceptibility signals of these two records in this interval (Supplementary Fig. 1) strongly substantiates our preferred correlating lines (Fig. 5). The second instance consists of the correlation between the stratigraphic levels at 18 and 21.5 m in Fuhe (in the *linguiformis* Zone) with the stratigraphic levels at 355.5 and 359 m at Section C (in the *rhenana* Zone) (Figs. 5 and 6). Given the good chemostratigraphic control in this interval close to the LKW (Fig. 6), we consider it most likely that the *rhenana–linguiformis* boundary is identified too low within the Fuhe section. Once all correlating time-lines were established, we used those tie-points to transfer the astrochronology from Section C to the other sections, using the ages indicated in Figs. 5 and 6. The black ages on these figures represent the initial ages that we assigned to the correlating lines, based on the 405-kyr astrochronology of Section C. The green ages on these figures represent a refinement of the initial ages after our Monte-Carlo-based age modeling approach (see "Methods" and Supplementary Figs. 2–4 for more details).

## Discussion

The multitaper method (MTM) spectral plots of the MS records that were tied-in into the astrochronologic framework (Fig. 2) show multiple peaks that are associated with astronomical forcing parameters. The occurrence of elevated spectral power at the expected frequencies of 100-kyr eccentricity in the H-32 and CG-1 power spectra (Fig. 2), and obliquity in the H-32, CG-1,

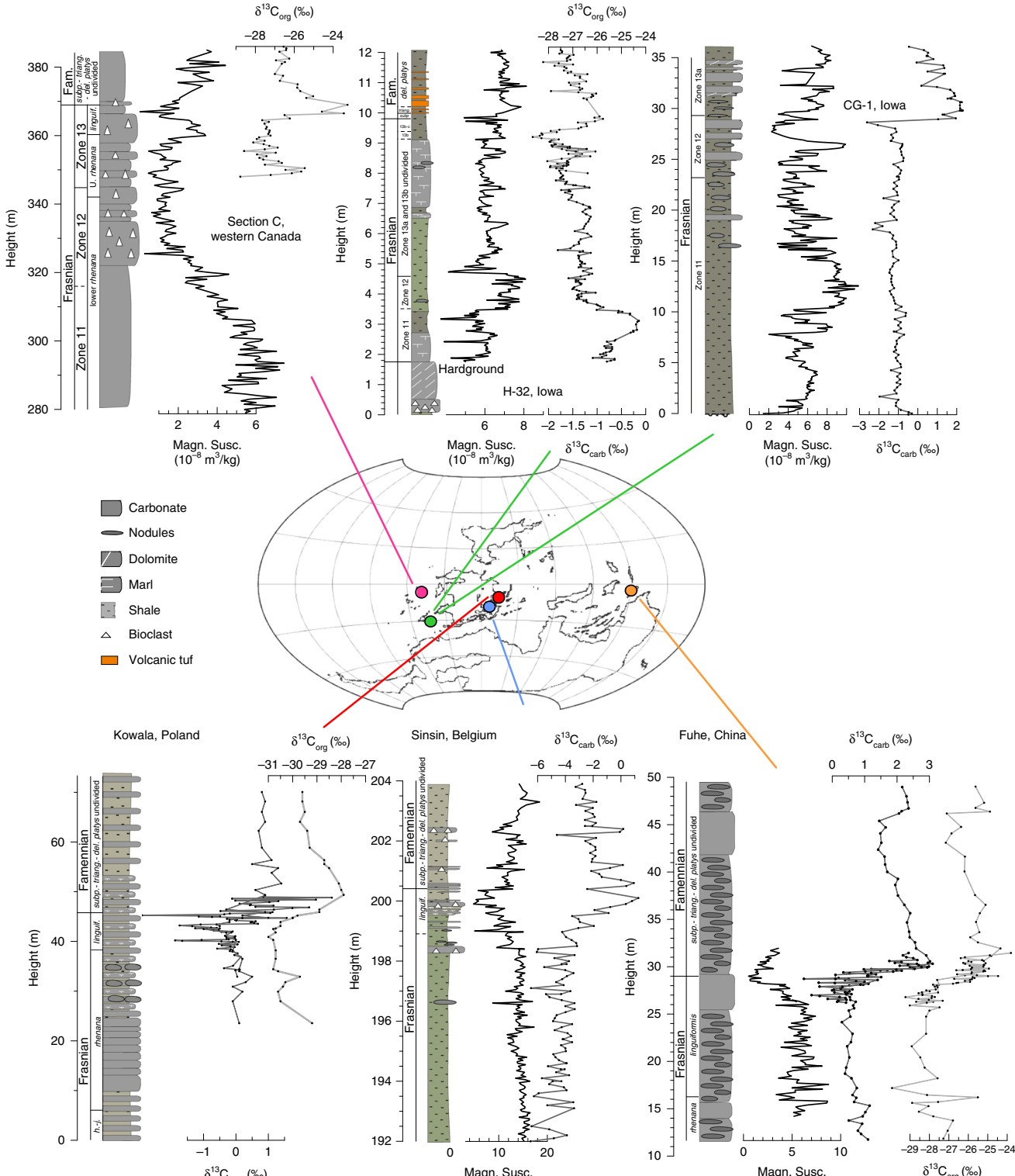

**Fig. 1** Frasnian–Famennian magnetic susceptibility and $\delta^{13}C_{carb}$ stratigraphies for six globally distributed sections. H-32, CG-1, Sinsin proxy records, and H-32 and CG-1 biostratigraphy were constructed in the framework of this study. The conodont biostratigraphy for the Sinsin section is from Sandberg et al.[58]. Section C (western Canada) magnetic susceptibility and biostratigraphy come from Whalen and Day[18] and the $\delta^{13}C_{carb}$ data from Whalen et al.[17]. High-resolution $\delta^{13}C$ data from Kowala are from De Vleeschouwer et al.[16], whereas the low-resolution $\delta^{13}C$ data and biostratigraphy come from Joachimski et al.[19]. Fuhe magnetic susceptibility data and high-resolution carbon isotope data are from Whalen et al.[3], low-resolution carbon isotope data are from Chen et al.[20], and biostratigraphy is from Chen et al.[21]. Throughout the figure, carbon isotope records in dark gray are $\delta^{13}C_{carb}$ and records in light gray are $\delta^{13}C_{org}$. The Late Devonian paleogeographic map comes from De Vleeschouwer et al.[34] and is inspired on reconstructions by Blakey[67]. Fam. Famennian, *h.-j. hassi-jamieae,* U. upper, *linguif. linguiformis, subp. subperlobata, triang. triangularis, del. platys delicatula platys*

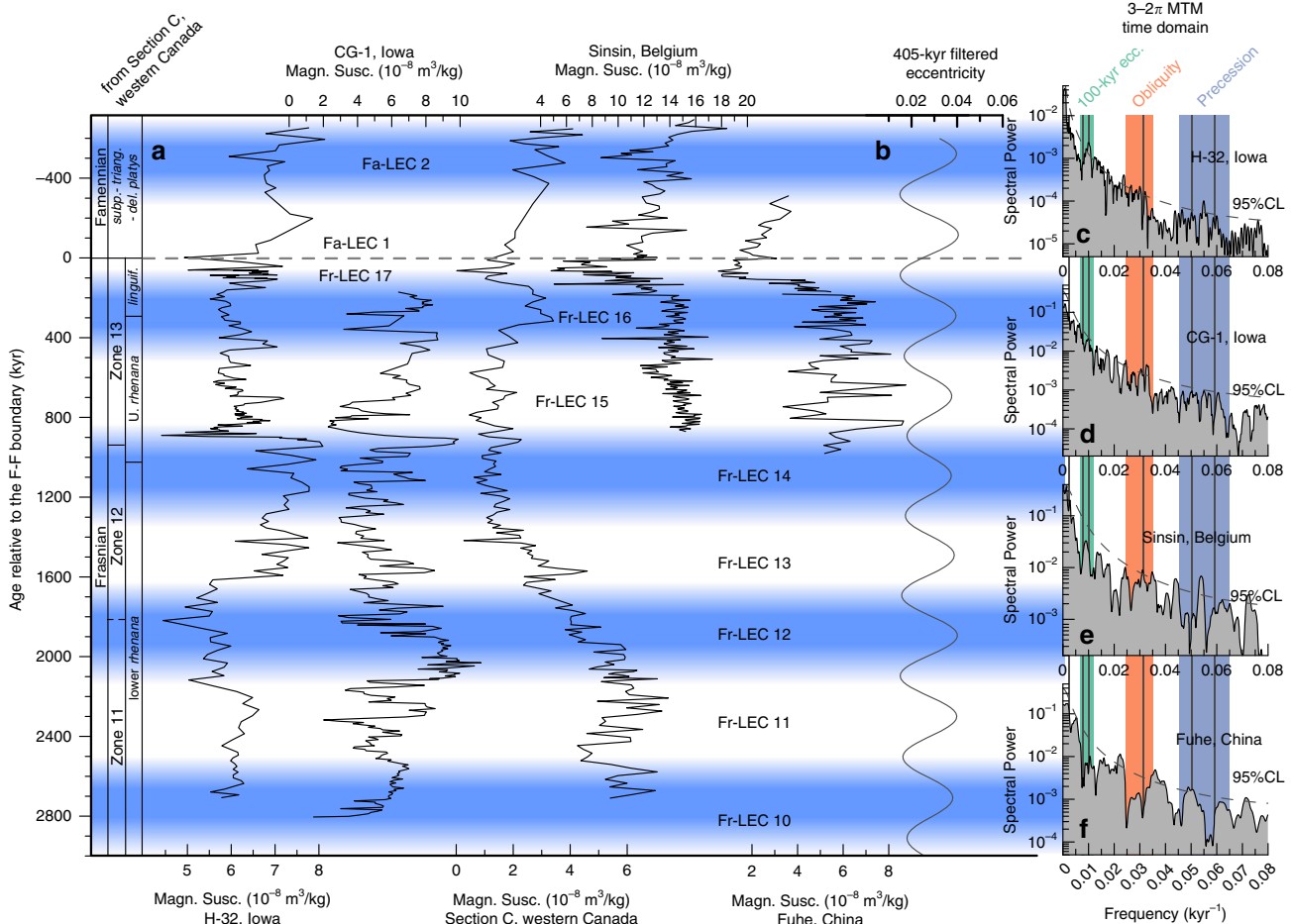

**Fig. 2** Late Frasnian–earliest Famennian magnetic susceptiblity data from globally distributed sections along a common astronomically constrained relative timescale. **a** The H-32, CG-1, Sinsin, and Fuhe records have been tied-in into the Frasnian astrochronologic framework for Section C (western Canada) and Kowala (Poland), based on distinct features in magnetic susceptibility and carbon isotope geochemistry (Figs. 5 and 6). The Frasnian astrochronogic framework counts 16.5 long 405-kyr eccentricity cycles (Fr-LEC)[15–17]. The first Famennian 405-kyr cycle (Fa-LEC 1) is the same cycle as Fr-LEC 17. **b** The 405-kyr eccentricity filter is derived from the La2004 solution[68], and plotted according to the phase relationships discussed in this study. **c–f** Multitaper method (MTM) power spectra of the tied-in records show peaks at or close to the expected frequencies of 100-kyr eccentricity (ecc.), obliquity and precession. Vertical lines across power spectra indicate the expected astronomical frequencies at 375 Ma, as calculated by Berger et al.[26]. linguif. linguiformis, subp. subperlobata, triang. triangularis, del. platys delicatula platys

and Sinsin power spectra adds confidence to the global astro-chronology. In order to subdue the risk for circular reasoning, we checked whether the astronomical imprint could also be observed when carrying out spectral analysis in the depth domain. Evolutive harmonic analysis (EHA) of the magnetic susceptibility depth-series of the H-32, CG-1, Fuhe, and Sinsin sections reveals that the different astronomical components can be traced in the depth domain (Supplementary Fig. 5). This implies that the astronomical imprint has not been introduced by the age modeling strategy adopted in this paper. The Fuhe section power spectrum is characterized by a significant spectral peak at 0.036 cycles/kyr (28 kyr period), which is slightly offset in comparison with the expected Devonian obliquity frequency of 0.031 cycles/kyr[26] (32 kyr period), as observed at H-32, CG-1 and Sinsin. Yet, we associate this peak in the Fuhe spectrum to variations in the obliquity of the Earth's rotational axis, implying that we observe obliquity-related peaks in all four spectra. This is remarkable given the (sub)tropical paleolatitudes of the studied sections, where the influence of obliquity on insolation is minor. A clear obliquity signal in (sub)tropical records is often interpreted as an indication of a climatic teleconnection between high and low latitudes[14], for example through the remote influence of waxing

and waning high-latitude ice-sheets. However, recent climate modeling results have demonstrated that tropical atmospheric circulation is highly susceptible to changes in the cross-equatorial insolation gradient[27]. This gradient is strongly influenced by obliquity and thus may explain a (sub)tropical obliquity imprint without invoking high-latitude ice sheets. In other words, the existence of continental ice sheets during the Frasnian remains an open question[28].

The carbon isotope stratigraphy in Fig. 3 places the LKW halfway Fr-LEC 15, while the UKW corresponds to Fr-LEC 17 (which is the same cycle as Famennian long eccentricity cycle 1 (Fa-LEC 1)). The LKW is a relatively short-lasting event, with a total duration of about 200 kyr. The onset of the positive LKW $\delta^{13}C$ excursion occurs ~800 kyr before the F–F boundary, while the end of the excursion falls 500–600 kyr prior to the F–F boundary. The peak of the LKW-positive $\delta^{13}C$ excursion corresponds to the maximum of Fr-LEC 15. The UKW is significantly more protracted, compared to the LKW. The onset of the UKW $\delta^{13}C$ excursion occurs in the lowermost part of Fr-LEC 17, about 150 kyr prior to the F–F boundary. The UKW excursion spans this entire 405-kyr cycle (Fr-LEC 17, Fa-LEC 1), with $\delta^{13}C$ values stabilizing ~400 kyr after the F–F boundary (Fig. 3). The LKW

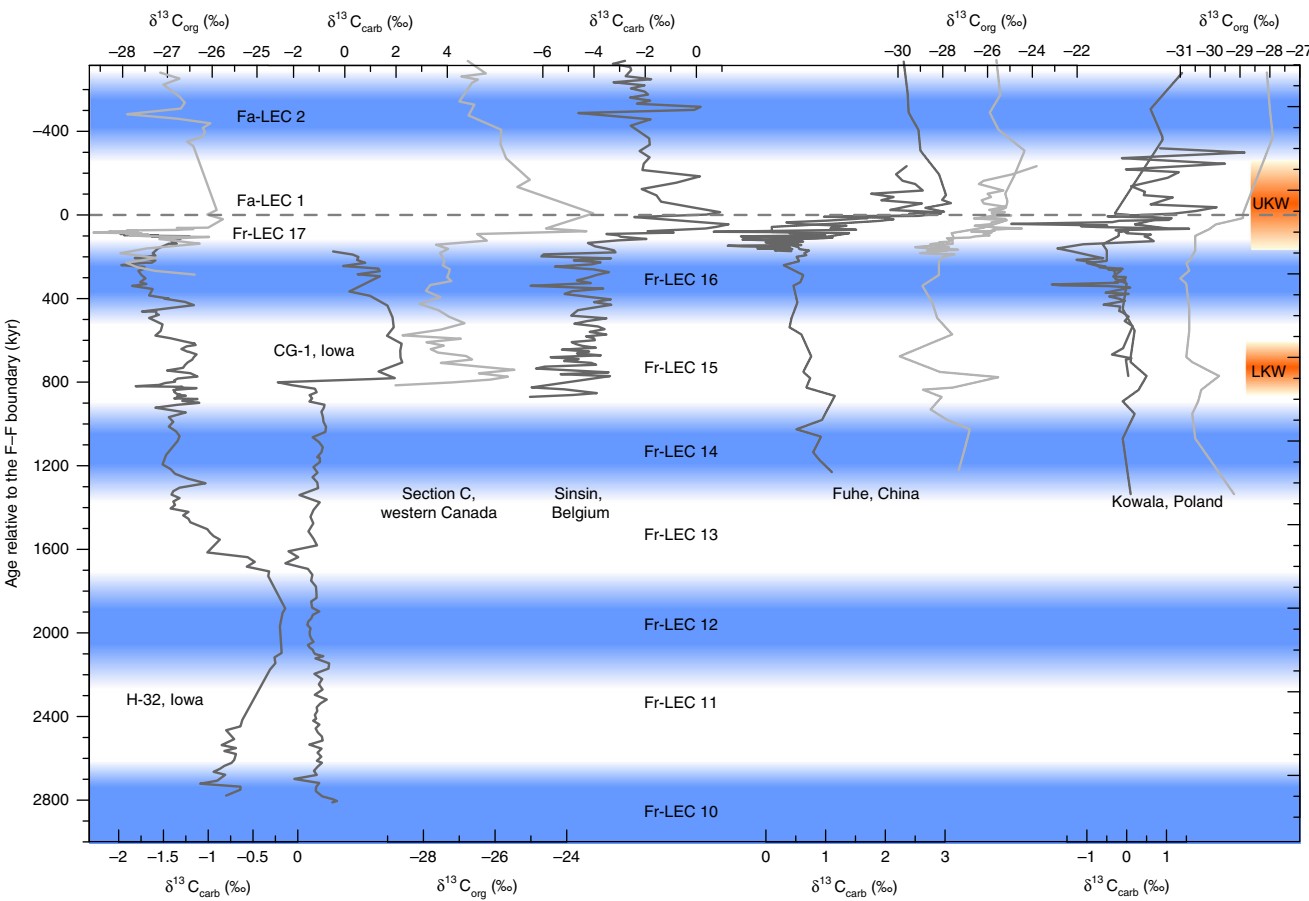

**Fig. 3** Late Frasnian–earliest Famennian carbon isotope geochemistry from globally distributed sections along a common astronomically constrained relative timescale. The H-32, CG-1, Sinsin, and Fuhe records have been tied-in into the Frasnian astrochronologic framework for Section C (western Canada) and Kowala (Poland), based on distinct features in magnetic susceptibility and carbon isotope geochemistry (Figs. 5 and 6). The Frasnian astrochronogic framework counts 16.5 long 405-kyr eccentricity cycles (Fr-LEC)[15-17]. The first Famennian 405-kyr cycle (Fa-LEC 1) is the same cycle as Fr-LEC 17. UKW upper Kellwasser, LKW lower Kellwasser

and UKW thus both start close to a 405-kyr eccentricity cycle minimum, reaching their peak in the subsequent 405-kyr maximum. In terms of sea level, the LKW started during the transgressive systems trackt of transgressive–regressive (T–R) cycle IId-2[17], and the UKW corresponds to the latest Frasnian–earliest Famennian sea-level lowstand and transgression of T–R cycle IId-3[29–32]. Maximum transgression is thus associated with maximum eccentricity during both Kellwasser events[33]. This is in accordance with numerical Late Devonian climate modeling, suggesting that high eccentricity configurations favored relatively warm and wet global climates[34], and represents the same phase relationship between eccentricity and global climate as during the Cenozoic[35]. The timing of the Kellwasser events with respect to eccentricity bears a remarkable resemblance to the timing of Ocean Anoxic Event-2 (OAE-2) just prior to the Cenomanian–Turonian boundary. Indeed, Batenburg et al.[36] place the base of the Livello Bonarelli (the expression of OAE-2 in the Umbria-Marche Basin, Italy) at the first short-eccentricity maximum after a 405-kyr minimum.

Having constructed an astronomically constrained relative timescale for the F–F interval, it is possible to evaluate the changing imprint of astronomical forcing across the UKW. Because of the similarity in timing of the UKW and OAE-2 described above, our assessment focuses on obliquity and follows the methodology introduced by Meyers et al.[37] for OAE-2. The high-resolution $\delta^{13}C_{carb}$ series from Kowala and Fuhe are best

suited for this analysis, given that they possess sufficient resolution (~10 kyr and ~6 kyr, respectively) to evaluate the imprint of obliquity. We perform EHA spectral analyses on the astronomically calibrated $\delta^{13}C_{carb}$ series, followed by integration of the spectrum over the obliquity band (0.015–0.04 cycles/kyr; Fig. 4). Maximum obliquity power occurs a few tens of thousands of years before the F–F boundary, both in the Kowala and in the Fuhe section. The marked increase in the strength of the obliquity signal at the time of global carbon cycle perturbation and organic carbon accumulation is thus an additional similarity between OAE-2 and the UKW. The strong obliquity signal during the earliest phase of the UKW could be interpreted as the influence of high-latitude climate processes on the paleoenvironmental setting of the studied (sub)tropical sections. However, this is rather unlikely given the greenhouse state of the climate system at the time of the F–F transition. Alternatively, a strong obliquity signal can be associated with a long-term minimum in eccentricity. We consider this interpretation to be more plausible, given that the obliquity imprint at Kowala reveals a notable increase, at the eccentricity minimum that separates Fr-LEC 15 and Fr-LEC 16, as well as at the eccentricity minimum separating Fr-LEC 16 and 17. The latter increase in obliquity power is associated with the onset of the UKW. Interestingly, several recent studies pointed to the link between long-term eccentricity forcing on the one hand and the Myr-scale behavior of the Mesozoic carbon cycle on the other hand[36,38–41]. These authors all associate positive excursions

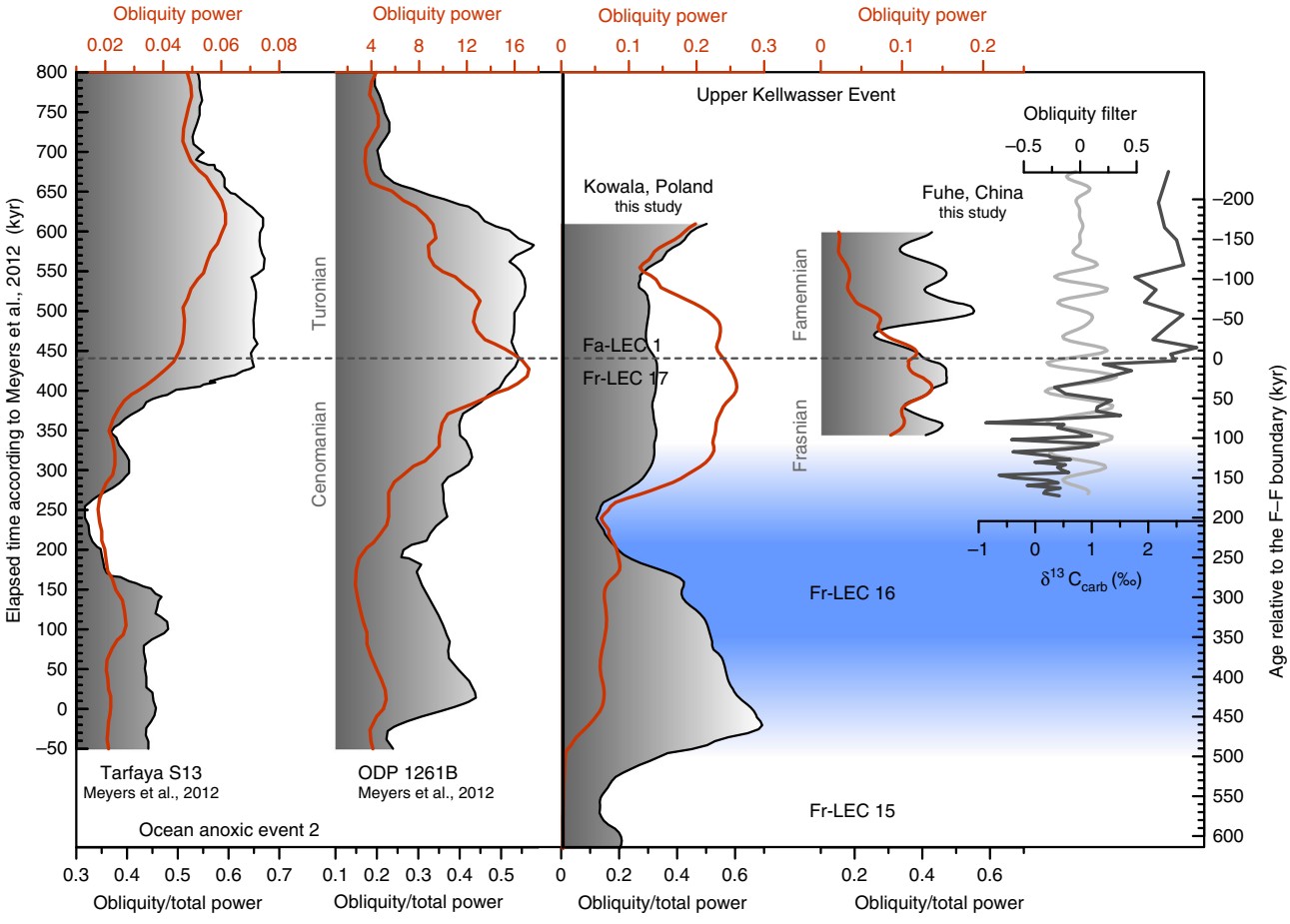

**Fig. 4** The changing imprint of obliquity forcing across OAE-2 and the UKW. Red lines indicate total power in the obliquity band (0.015–0.04 per kyr), and the black lines indicate the fraction of the power in the obliquity band, as compared to total power at frequencies less than 0.1 per kyr. All analyses have been conducted using a 3–2π MTM power spectra and a 300-kyr moving window (150-kyr window for the Fuhe section). The Fuhe δ13Ccarb time series and its obliquity-filter visualize the notable imprint of obliquity on the course of the UKE positive δ13Ccarb excursion

in δ13C with so-called "nodes" in eccentricity, i.e., minima of the 2.4 Myr or 9 Myr eccentricity cycle. During such a node, eccentricity is low for a prolonged period of time, such that the influence of precession on insolation is dampened and seasonal extremes in insolation are avoided. In other words, a strong obliquity signal is in agreement with an eccentricity node, being the only remaining astronomical parameter with a significant influence on insolation. In this scenario, the onset of the UKW δ13C excursion would not only correspond to the 405-kyr eccentricity minimum between Fr-LEC 16 and Fr-LEC 17, but also to a minimum of the long-term 2.4-Myr eccentricity cycle. During this period of low eccentricity, organic carbon on land could accumulate due to the avoidance of climates with extreme seasonality, causing marine δ13C to rise[36,39,42] at the beat of obliquity. A few tens of thousands of years later, eccentricity increased rapidly and reaches its subsequent 405-kyr maximum (Fr-LEC 17, Fa-LEC 1). This sequence of astronomical configurations supposedly contributed to a rapid warming of global climate as well as to an intensification of the hydrological cycle[34]. Under such climatic conditions, sea level rise and increased weathering are instigated and likely contributed to the eutrophication of shallow seas[43], and thus to the widespread deposition of the organic-rich UKW. Yet, astronomical climate forcing is not ultimately responsible for the Late Devonian mass extinction. During the Devonian, the global carbon cycle was in a vigorous state, with enhanced silicate weathering under warm and humid conditions, and increased carbon burial through a productivity-

hypoxia-nutrient recycling feedback loop[44–47]. This state of vigorous carbon cycling made the Devonian quite vulnerable for ocean anoxia and mass extinction[48]. The ultimate cause for the Late Devonian mass extinction thus probably lies with a unique convergence of processes invigorating the global carbon cycle (e.g., evolution of land plants[43], volcanism[49,50], tectonic processes[51–53]), but the exact timing of the outbreak of the LKW and UKW seems to be linked to a particular succession of astronomical configurations.

In conclusion, the construction of a global astrochronology across the F–F boundary interval reveals the steering role of astronomical climate forcing during one of the most far-reaching biotic events in Earth's history. Intriguing obliquity-signatures characterize the Late Devonian UKW event, as well as the Cretaceous OAE-2. These two greenhouse worlds are separated by ~200 million years. Nonetheless, astronomical "*Milankovitch*" forcing seems to have influenced global carbon cycle dynamics in a similar way during both periods. In both cases, long-term minima in eccentricity ("nodes") coincide with major perturbations of the global carbon cycle and increased extinction rates, with devastating effects during the Late Devonian extinction event.

## Methods
**Geologic settings.** The Kowala section (50.79° N, 20.57° E) was deposited in the intrashelf Chêciny-Zbrza Basin. The F–F interval at Kowala consists of thin-bedded marls with intercalated limestone beds and cherty limestone strata with marly

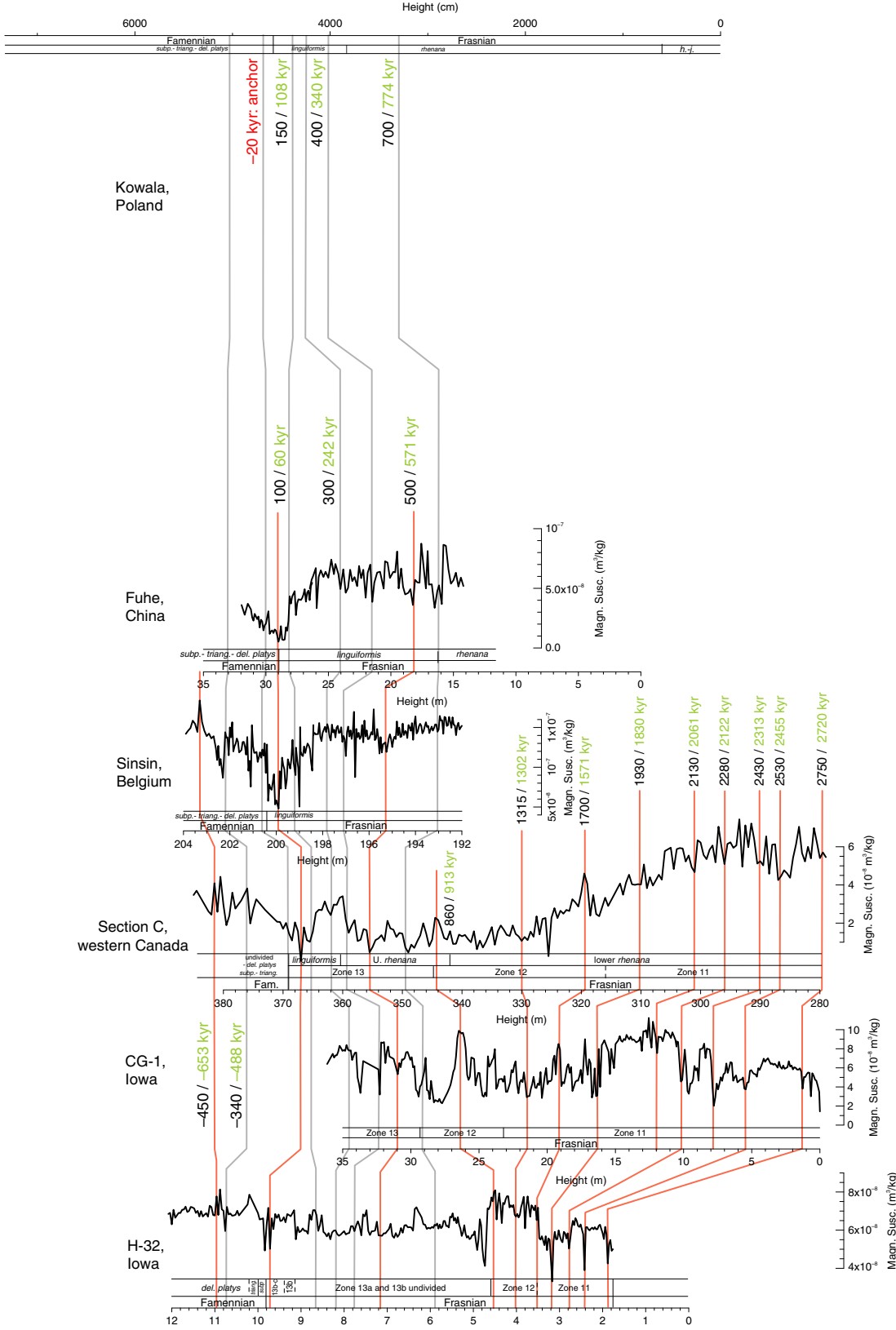

**Fig. 5** Magnetic susceptibility correlation. Tie-points between H-32, CG-1, Section C, Sinsin, and Fuhe were obtained by visually correlating distinct features in magnetic susceptibility (red ties). The gray ties were established by correlating distinct features in carbon isotope geochemistry (Fig. 6). We construct a common floating age model for all sections by assigning an age relative to the F–F boundary to each tie-point, according to the existing astrochronologic framework of Section C and Kowala[15, 16]. These ages are listed in black. The tie-point ages listed in green indicate the "optimized" tie-point ages, after 5000 Monte Carlo simulations ("Methods"). The latter tie-point ages are used for the depth-to-time conversion used in Figs. 2 and 3. Fam. Famennian, *h.-j. hassi-jamieae*, U. upper, *linguif. linguiformis*, *subp. subperlobata*, *triang. triangularis*, *del. platys delicatula platys*

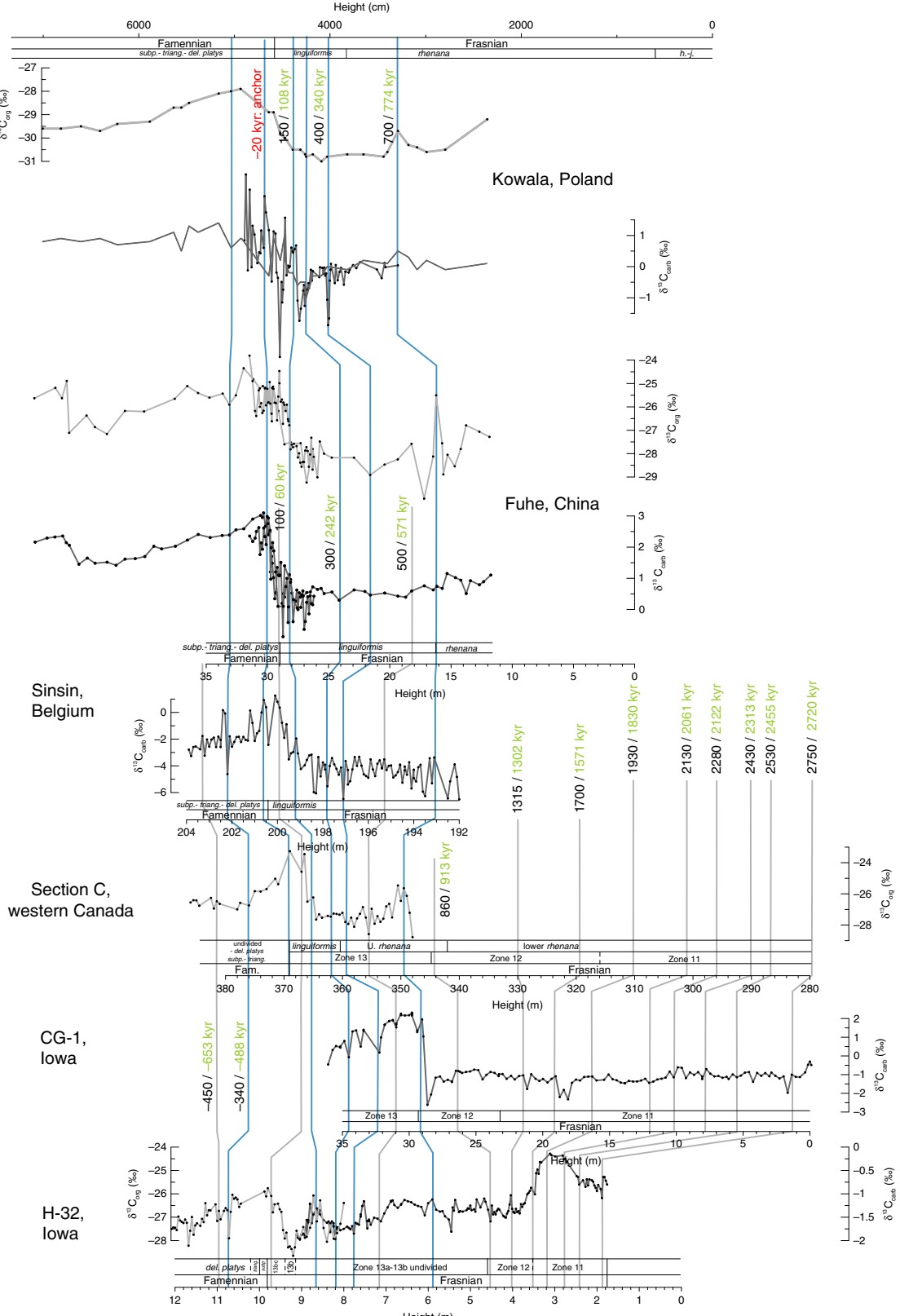

**Fig. 6** Carbon isotope correlation. Tie-points between H-32, CG-1, Section C, Sinsin, Fuhe, and Kowala were obtained by visually correlating distinct features in carbon isotope geochemistry (blue ties). The gray ties were established by correlating distinct features in magnetic susceptibility (Fig. 5). We construct a common floating age model for all sections by assigning an age relative to the F–F boundary to each tie-point, according to the existing astrochronologic framework of Section C and Kowala[15, 17]. These ages are listed in black. The tie-point ages listed in green indicate the "optimized" tie-point ages, after 1000 Monte Carlo simulations ("Methods"). The latter tie-point ages are used for the depth-to-time conversion used in Figs. 2 and 3. Fam. Famennian, h.-j. hassi-jamieae, U. upper, linguif. linguiformis, subp. subperlobata, triang. triangularis, del. platys delicatula platys

intercalations. The LKW and UKW horizons are marked by two increases in total organic carbon (TOC) content, but the two black shale horizons that typify most F–F sections in Central Europe are absent here[19]. The Kowala section holds a previously published astrochronology, which is based on the stacking pattern of the marl-limestone alternations[16]. Both Kellwasser-equivalent TOC maxima, as well as four younger black shale horizons in the Famennian part of the section, were each deposited during just a fraction of one 405-kyr cycle according to that chronology. This finding was later confirmed for the Hangenberg shale, through high-precision U–Pb dating[54]. For the F–F interval, high-resolution $\delta^{13}C_{carb}$ ($N=77$) and low-resolution $\delta^{13}C_{carb}$ data ($N=34$) are available from De Vleeschouwer et al.[16] and low-resolution $\delta^{13}C_{carb}$ ($N=38$) data are available from Joachimski et al.[19]. To our knowledge, no magnetic susceptibility series exists for the F–F interval of the Kowala section.

Section C (53.17° N, 118.18° W) is located along the southeast margin of the Ancient Wall platform in the western part of the Western Canada Sedimentary Basin[55]. The section records upper-slope and prograding outer-ramp facies. The base of Section C is late Frasnian in age and consists of clay- and locally silt-rich shales of the Mount Hawk Formation, which is characterized by high magnetic susceptibility values throughout the basin[18,55]. A sea-level deepening event in the very late Frasnian resulted in the progradation of ramp facies and low magnetic susceptibility values at Section C, reflecting the increased carbonate input. The early Famennian Sassenach Formation records an influx of siliciclastics into the basin and related higher magnetic susceptibility values. From Section C, 50-cm spaced magnetic susceptibility data ($N=365$)[18] and $\delta^{13}C_{org}$ ($N=52$)[17] are available.

During the Late Devonian, the Fuhe section (24.99° N, 110.41° E) was located in a tectonically active basin, surrounded by carbonate platforms. Pelagic nodular limestones intercalated with calciturbidites and debris flow deposits dominate the F–F sequence. Chen and Tucker[56] interpreted lithological cycles in the Fuhe section as the expression of precession cycles, stacked into meter-scale cycle-sets interpreted as 100-kyr eccentricity cycles. The UKW event is clearly marked by a +3‰ excursion in the high-resolution 10-cm spaced $\delta^{13}C_{carb}$ ($N=54$) and $\delta^{13}C_{org}$ ($N=52$) records[3]. The LKW event occurs in the lower part of the section, in the upper *rhenana* conodont zone, and is characterized by a positive excursion in the low-resolution $\delta^{13}C_{carb}$ ($N=73$) and $\delta^{13}C_{org}$ ($N=54$)[20] (Fig. 1).

The Sinsin roadcut section (50.28° N, 5.24° E) consists of shales, marls, and (nodular) limestones, deposited on the continental shelf below fair-weather wave base[57]. In the framework of this study, we sampled the section at 5-cm resolution for magnetic susceptibility ($N=223$) and at 10-cm resolution for $\delta^{13}C_{carb}$ ($N=116$). The carbon isotope chemostratigraphy exhibits a gradual shift towards more positive $\delta^{13}C_{carb}$ values (Fig. 1). This +3‰ excursion concurs with the upper *linguiformis* conodont Zone[58] and is therefore interpreted as the expression of the UKW. In the same interval as the positive $\delta^{13}C_{carb}$ excursion, TOC values peak up to 0.2 wt% in a 30-cm thick interval directly underlying the F–F boundary[59]. A previously published cyclostratigraphy does not exist for the Sinsin section. Kaiho et al.[59] demonstrated the imprint of astronomical forcing in the Sinsin section through wavelet analysis, but did not delineate individual cycles.

The CG-1 core (42.93° N, 93.08° W) was deposited in a shelf margin environment and consists of fossiliferous shales and argillaceous limestones of the Lime Creek Formation. In the framework of this study, we sampled CG-1 at an average resolution of 11 cm for magnetic susceptibility ($N=329$) and 26 cm for $\delta^{13}C_{carb}$ ($N=137$). The H-32 core (40.47° N, 91.47° W) is the basinal, deep-water time-equivalent section of CG-1 and consists of the dark-colored Sweetland Creek shale. The overlying Grassy Creek shale contains seven layers of volcanic ash, none of which yielded datable zircons. We sampled H-32 at 3 or 4-cm intervals for magnetic susceptibility ($N=294$) and at irregular intervals for $\delta^{13}C_{carb}$ ($N=165$) and $\delta^{13}C_{org}$ ($N=70$). The carbon isotope chemostratigraphy for both sections exhibits distinct positive $\delta^{13}C$ excursions (Fig. 1). The +3‰ excursion in the CG1 core begins within the upper part of Frasnian Zone 12. We thus interpret this excursion as the expression of the LKW. At H-32, a +2‰ $\delta^{13}C_{org}$ excursion spans Zones 13b and 13c. We interpret this excursion as the UKW as the F–F boundary occurs at the base of the first ash bed just below 1000 cm. Between 800 and 900 cm, a smaller positive excursion is observed in both $\delta^{13}C_{carb}$ and $\delta^{13}C_{org}$, which we interpret as the LKW. The H-32 and Cerro Gordo Project Hole 1 (CG-1) drill cores have never been studied in terms of astronomical forcing before.

**Magnetic susceptibility.** Magnetic susceptiblity measurements for the Fuhe and Sinsin sections were taken with a KLY-3S kappabridge at the University of Liège (Belgium). Samples from the CG-1 and H-32 cores were measured at the University of Wisconsin at Milwaukee (USA) with a MFK1 kappabridge. Both devices have a noise level $<2 \times 10^{-8}$ SI. Each magnetic susceptibility data point is the average of three measurements and has been corrected for sample mass. Samples typically have a mass larger than 10 g and were weighed with a precision of 0.01 g.

**Carbon isotope stratigraphy.** Bulk carbonate $\delta^{13}C$ from the H-32, CG-1 and Sinsin sections and $\delta^{13}C_{org}$ from H-32 were measured at the Vrije Universiteit Brussel (Belgium). $\delta^{13}C_{carb}$ was measured using a NuPerspective IRMS interfaced with a NuCarb automated carbonate device. We use the internal NCM standard calibrated against NBS19 (+2.09‰). The long-term standard deviation of the internal standard is 0.05‰ for $\delta^{13}C_{carb}$. Powdered samples for $\delta^{13}C_{org}$ were

weighed (~10 mg) in silver capsules. The samples were decarbonated by adding 5% HCl in steps of 4 h and reaction at 50 °C, till no further reaction occurred. The $\delta^{13}C_{org}$ was measured using a Thermo1112 flash elemental analyzer coupled via a Con-Flo III interface to a Thermo DeltaV isotope ratio mass spectrometer (IRMS). The international reference standard IAEA-CH-6 (−10.449‰) was used for calibration ($1\sigma \leq 0.2$‰).

**Conodont faunas.** The H-32 and CG-1 zonal nomenclature, presented in this study, follows the newest Frasnian[60] and Famennian[61] standard zonations.

We took 118 samples for biostratigraphy throughout the CG-1 core. All samples are typical of the Frasnian *Polygnathus* biofacies. Samples from the lower 23 m of the Juniper Hill and lower Cerro Gordo members include *Palmatolepis semichatovae* and *Pa. kireevae* correlated with Frasnian Zone 11[62,63]. The first appearance of *Polygnathus planarius* is interpreted as marking the base of Zone 12 based on its known range in western Canada. The occurrence of *Polygnathus samueli* high in the Cerro Gordo Member, just below the onset of the LKW $\delta^{13}C$ excursion, is high within Frasnian Zone 12. The joint occurrence of *Ancyrognathus asymmetricus* and *Icriodus alternatus* in the Owen Member is interpreted as Frasnian Zone 13a, as the latter first occurs very high in Zone 12 and ranges into Zones 13a and 13b. Stratigraphically, this level occurs above peak values of the LKW $\delta^{13}C$ excursion.

We took 54 samples for biostratigraphy throughout Core H-32. Most of the Sweetland Creek Shale at H-32 correlates with Frasnian Subzones 13a to 13b. The overlying Grassy Creek shale, just below the F–F boundary, corresponds to Frasnian Subzones 13b–13c (i.e., the *linguiformis* interval). The conodont yields of H-32 samples are generally very small or barren, but nevertheless, clear zonal indicators allow for the construction of the biostratigraphic scheme presented in Fig. 1. The Zone 11 and Zone 12 interval in the Sweetland Creek Shale is condensed in less than 3 meters. Most of the Sweetland Creek Shale at H-32 represents an expanded record of the LKW through the very Early Famennian.

**Age modeling and spectral analyses.** Tie-points between H-32, CG-1, Sinsin, Fuhe, Kowala, and Section C are obtained by visually correlating distinct features in magnetic susceptibility (red ties in Fig. 5) and carbon isotope geochemistry (blue ties in Fig. 6). In a first step, we assign an age relative to the F–F boundary to each of these tie-points according to the 405-kyr astrochronologic framework of Section C (Figure 7 in De Vleeschouwer et al.[15]). These ages are listed in black in Figs. 5 and 6. In a second step, we apply a Monte Carlo procedure to distort time-differences between consecutive tie-points. The goal of this procedure is to slightly modify the tie-point ages, so to get a better expression of the astronomical frequencies in the power spectra of the F–F proxy-series. We run the Monte Carlo procedure 5000 times in our search for the best astronomical fit. The tie-point ages for the best-fitting Monte Carlo simulation are listed in green in Figs. 5 and 6. The fit of each Monte Carlo simulation is evaluated as follows: we compute the power spectra for all susceptibility time series and consider the spectral power in the 100-kyr eccentricity (0.008–0.013 cycles/kyr), obliquity (0.025–0.035 cycles/kyr) and precession (0.045–0.065 cycles/kyr) bands. A good astronomical fit exists when high spectral power occurs in each of these bands. We measure this fit by subtracting the maximum AR1 confidence level occurring within each band from 100%. Low percentages thus represent high spectral power within the considered frequency band. Additionally, we measure the misfit between the frequency of maximum AR1 confidence level, and the expected orbital frequency according to Berger et al.[26] (at 375 Ma: 0.0105 per kyr for 100-kyr eccentricity, 0.031 per kyr for obliquity and 0.055 per kyr for precession). This frequency misfit is also expressed as a percentage. Finally, we measure the total astronomical fit of the Monte Carlo simulation in question by averaging the spectral power and misfit measures among all frequency bands and among all sections. Our age modeling strategy is illustrated in detail using three distorted eccentricity-tilt-precession (ETP) series in Supplementary Figs. 3–5.

All spectral analyses in this study were carried out using the MTM with three $2\pi$-tapers[64], as implemented in the R-package "astrochron"[65]. The confidence levels were calculated applying the conventional AR1 method for estimating the red noise spectrum[66]. The R-code used for age modeling and spectral analyses is available through GitHub ("Code availability").

**Code availability.** Software S1. R-script for the illustration of the age modeling strategy adopted in this study, using distorted eccentricity-tilt-precession (ETP) series is available at https://github.com/dadevlee/Late_Devonian. Software S2. R-script for age modeling of Late Devonian data is available at https://github.com/dadevlee/Late_Devonian. Software S3. R-script for spectral analyses and the generation of the data in Figs. 2, 3 and 4 is available at https://github.com/dadevlee/Late_Devonian.

**Data availability.** Magnetic susceptibility and carbon isotope series are available from https://github.com/dadevlee/Late_Devonian/tree/master/Data_depth_domain and https://doi.pangaea.de/10.1594/PANGAEA.882366.

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

## Acknowledgements

This research was supported by PhD fellowships from the Research Foundation Flanders (FWO) to D.D.V. (1148713N and V421513N) and M.S. (FWOTM782). We thank the Académie royale de Belgique for financial support (Prix Annuel en Géologie 2014 awarded to D.D.V.). D.D.V. is a postdoctoral researcher in the ERC Consolidator Grant "EarthSequencing" awarded to Heiko Pälike. A.-C.D.S. acknowledges the Netherlands Organization for Scientific Research (NWO, grant WE.210012.1). D.C. and Z.G. were supported by the National Natural Science Foundation of China (Grants 41072079 and 41290260). P.C. thanks the FWO Hercules program and the VUB strategic research program. This paper is a contribution to IGCP-652 "Reading geologic time in Paleozoic sedimentary rock".

## Author contributions

D.D.V., A.-C.D.S., and M.S. conceived the study and carried out the research; D.D.V. wrote the paper; D.D.V., A.-C.D.S., M.S., D.C., J.E.D., M.T.W., and Z.G. analyzed data; P. C., J.E.D., and M.T.W. contributed to interpretation of data.

## Additional information

**Competing interests:** The authors declare no competing financial interests.

