## [Peer review file · Nature Communications]

Reviewers' comments:

Reviewer #1 (Remarks to the Author):

This is my review of the manuscript "Obliquity-pacing of the Late Devonian mass extinction event" submitted by De Vleeschouwer et al. to Nature Communications. In this contribution, the authors attempt to shed light on the pacing of paleoenvironmental events which accompanied one of the five mass extinctions at the Frasnian-Famennian (F-F) boundary. This work is firstly based on new biostratigraphy, magnetic susceptibility and carbon isotope data from four sections (or cores) across the Frasnian–Famennian boundary. These proxies are used to correlate each section to two others whose the cyclostratigraphical calibration was already published. Hence, relative depths are converted in numerical ages (based on the succession of 405 kyr eccentricity cycles), allowing the comparison of remote paleoenvironmental signals in a same temporal referential. Finally, the authors calculate the obliquity power recorded in the $\delta^{13}\text{C}$ signals of two sections and demonstrate that this orbital cycle was preponderant during the Kellwasser events responsible for the Late Devonian extinction.

After several readings of the manuscript, I have a mixed feeling about this paper. On the one hand, it is well written, the new data are interesting, the Monte-Carlo intercalibration method of temporal scales is great, and the conclusions promising for our understanding of the Late Devonian events. On the other, the methodological part is confusing (with some circular reasoning), some parts presented as results are already published and bring length to the text, and I do not understand why the authors detailed and intercalibrated six sections if only two of them (whose one already published) are used to support the conclusion about the obliquity pacing of Late Devonian events. Moreover, I think that the title is not appropriate regarding the discussion. Indeed, the authors explain that the obliquity signal was well expressed in the sedimentary series because the eccentricity power was low. As the coeval rises of $\delta^{13}\text{C}$ (and related anoxic events) are only explained by low eccentricity parameters, I conclude that the title should start with "Eccentricity pacing...". Finally, I am convinced by the data that an orbital factor likely paced the Devonian anoxic events (and organic deposits) but if I come back to the title once again, have we quantified evidences for cyclic fluctuations in diversity levels across the F-F boundary? The authors never clearly explain the orbital link with the dynamics of biodiversity crisis. I mean that orbital configurations with low eccentricity parameters are cyclic and occurred before and after the F-F boundary. So why this orbital parameter triggered lethal conditions only during these two intervals? These points are especially lacking in the text, and could bring more interest to the study, especially for a publication in Nature Communications.

For all these reasons, I conclude that the paper is interesting but cannot published in Nature Communications in its present form. Major revisions are needed to clarify some methodological aspects and bring coherence between title and discussion. Some of these recommendations are further explained below:

INTRODUCTION: The introduction provides a good explanation of the problem. But from lines 49 to 57, I think that the authors need to tell what is really new and what is already published by the authors. A brief description (in one or two sentences) of the orbital calibration is also needed because the result parts in difficult to follow without methodological keys.

Line 42: The authors mention "environmental changes responsible for the mass extinction". Except anoxia and organic matter deposition, what are there?

METHODS: I acknowledge the authors in their effort to propose a global temporal calibration of sections. However, some points may be further discussed or explained:

- First, basic information regarding sampling, number of new data, and outcrops are completely lacking. This is one important aspect of the work, which deserves to be mentioned. Moreover, I suggest moving facies and paleoenvironmental descriptions of cores and sections in the Methods, as the sedimentology of most of them is already described in publications.
- The authors correlate the outcrops and cores by using magnetic susceptibility data and $\delta^{13}\text{C}$ data from remote domains. Correlation lines are shown on supplementary figures but I wonder how some correlation lines are drawn. For me, magnetic susceptibility signals are rarely global and may present local variations or peaks (highly dependent on sampling resolution and facies). For example, the authors mention that debris flows are present in the Fuhe section... How to deal with? Moreover, the correlation of MS peaks is sometimes very subjective on figures... I suggest to the authors: 1) to add biostratigraphical scheme of each section to have a better appraisal of correlations; 2) to better explain how MS and $\delta^{13}\text{C}$ peaks are correlated; 3) to further explain how the Monte Carlo approach can help to measure errors in the position of tie-points; 4) to discuss how difference in the sampling effort of section can affect correlations.
- I understand that the authors apply the well-defined orbital calibration of the Canadian and Polish sections to the other cores to define a common temporal scheme. However, I was surprised that they never analyzed the raw MS patterns of new studied sections to show that the 405 kyr cycles used for relative dating are well recorded. Instead, they analyzed the MTM power spectra of time-corrected MS records to show that expected eccentricity, obliquity, and precession frequencies are well present. Is it not a circular reasoning? We could suggest that a previous time calibration (by 405 Myr cycles) artificially introduce expected high-frequency peaks, isn't it? The authors have to better explain these points in the method and show the spectral analyses of raw MS data.
- On Figure 2, I remark that, contrarily to what explained in the text, the expected eccentricity, obliquity, and precession cycles are sometimes below the 95% confidence interval (in Sinsin for example). This is important to mention in the text and propose explanation (local paleoenvironmental influence?).
- The age modelling method described at lines 253-265 is very original. However, I think that a supplementary figure with examples would help to clarify.

RESULTS: In my opinion, the description of cores and sections is very long and sometimes boring since some results (of Polish and Canadian sections) are already published. I suggest shortening this part by further focusing on the new data (including biostratigraphy, MS data, and $\delta^{13}\text{C}$). Other points may be included in the Method.

L74: I do not think that the potassium content is useful as the authors never describe it or use it as correlation tool. Maybe to remove in the figure 1.

DISCUSSION:

L159-162: The link between eccentricity, sea level, and warming event is unclear. Please, better explain with bibliographical references showing this link.

L166: Is this hypothesis coherent with the recent papers of Eldrett et al. (2015) "Origin of limestone-marlstone cycles: Astronomic forcing of organic-rich sedimentary rocks from the Cenomanian to early Coniacian of the Cretaceous Western Interior Seaway, USA, EPSL

L183: It is disturbing to analyze the obliquity power along two sections to conclude that this is related to a low eccentricity... Is it possible to measure the eccentricity power for the 6 locations by focusing on eccentricity (<0.02) or precession frequencies? A test on the H-32 Iowa core would be maybe more relevant to show the patterns on the long term.

FIGURES:

The addition of 405kyr eccentricity filters on the right would be helpful to show when eccentricity is low or high (colours are not explained).

Reviewer #2 (Remarks to the Author):

Review of De Vleeschouwer et al #132062

Overall this is an important manuscript that deserves publication. The manuscript could benefit from some improvements in the presentation of language, particularly in the front half of the manuscript. Overall I have mostly linguistic suggestions, however, I think the authors can substantially improve the front of the manuscript. In particular, it currently is disjointed in the beginning. It almost reads more as a list of facts than as a coherent manuscript. The back half of the manuscript, once you get to the discussion reads much more comfortably. I would suggest either including much of the details about each specific section in the supplementary materials section, or figure out some other way to make this less like a Gregorian chant of information. The interesting bits are presented all at once in the end of the manuscript with little to no warning that there is going to be that much of interest. In short, there is nothing up front that begs the reader to keep reading. The authors can fix this quite easily I would imagine and presage where you are going with the front of the manuscript. Move much of the other information, which is too short to be terribly useful anyhow, to the supplementary sections. When you do that, it will also allow you to increase the amount of detail you can include about each section.

I strongly believe this manuscript should be published following these minor revisions. Below is a list of specific comments on the manuscript by line number.

Ln. 23: Throughout the manuscript, please be careful with chronostratigraphic vs geochronologic terms.

Because you are talking about stratigraphy here, you need to use Upper Devonian. Similarly, please

refrain from using the word 'period' colloquially. This word has a specific geologic meaning. I would suggest that you replace it with the word 'interval'.

Ln. 24: I would suggest you use 'orbitally-tuned' or 'orbitally-calibrated'

Ln 26: Unless instructed by the journal, it is not normal to put the $\delta^{13}C$ in italics. Please make it regular

text. Please fix this throughout the manuscript.

Ln 27: Again, don't use 'periods' here.

Ln. 40: Since you are talking about time here, please use Middle to Late Devonian.

Ln. 44: Please correct this to 'constraints on the calibration of the stage boundaries within the Devonian

System...

Ln. 44: Please remove 'therewith'...it is awkward.

Ln. 50: duration of an entire age...

Ln 60: We studied...this is past tense.

Ln. 64: replace 'passage' with 'interval'

Ln. 66: the word 'episodic' is problematic. Please remove.

Ln. 67: ...content, but the two black shale...

Ln. 96: A section cannot possess...Please rewrite this sentence.

Ln. 103: Again, do not anthropomorphize stratigraphic section...a core cannot hold anything.
Please

rewrite.

Ln. 112: Please remove 'which makes that'...Also, this is an example of the type of detail that belongs in

the supplementary section. This reference to meterage here comes out of nowhere and can be much better

dealt with more thoroughly in the supplementary materials section.

Ln. 133: ...ends halfway through Fr-LEC14

Ln. 137: please do not abbreviate here...what is resp.?

Ln. 152: Replace 'height'... to what are you referring? Stratigraphic height? Maximum isotope values?

Stratigraphic thickness?

Ln 176: ...few tens of thousands... Same again on line 197

Figures: I would prefer it if you included headings for each of your columns...Stage, Conodont Zone, etc...also, you need to include a legend for your stratigraphic columns.

Reviewer #3 (Remarks to the Author):

Vleeschouwer et al present logs of magnetic susceptibility (MS), $\delta^{13}\text{C}_{\text{org}}$, $\delta^{13}\text{C}_{\text{carb}}$ for six sections in five countries straddling the Frasnian/Famennian boundary. They focus on correlating the sections using biostratigraphy, the MS data and $\delta^{13}\text{C}$ combined with existing astrochronological scales in order to investigate the timing of the inferred global extinction event. The correlations presented are designed to demonstrate that there is common orbital-obliquity forcing of the changes in $\delta^{13}\text{C}$ and hence of extinction processes.

However, I find the correlations presented to be wholly unconvincing because: a) the authors have not respected the biostratigraphic constraints they provide in figure 1, b) correlation of magnetic susceptibility variations between basins is invalid since the sections contain entirely different lithologies and c) it is far from clear that the $\delta^{13}\text{C}$ data can be relied on for correlation globally except at the Frasnian/Famennian boundary. These are serious concerns that lead me to reject the paper. In more detail:

a) Biostratigraphic constraints not followed.

On lines 120 to 123 it is stated: "We tie the four sections (H-32, CG-1, Sinsin and Fuhe) into the common astrochronological framework of Section C and Kowala by correlating distinct features in magnetic susceptibility (red ties on Suppl. Fig. 1) and carbon isotope geochemistry (blue ties on Suppl. Fig. 2)." This approach makes sense because biostratigraphic boundaries are much more likely to approximate time lines than magnetic susceptibility or $\delta^{13}\text{C}$ variations.

However, comparing Fig. 1 with Fig. 2 it is clear that the authors have not stuck to their procedure. For example, the Zone 11/12 boundary of H-32 (Iowa) is located at Fr-LEC 11 in Fig. 2 whereas the same boundary at CG-1 (Iowa) occurs in Fr-LEC 14. Similarly the Zone 12/13 boundary of H-32 occurs in Fr-LEC 13 in Fig 2, but in Fr-LEC 15 for CG-1 and at a different time in Fr_LEC 15 for Section C (western Canada). Also the boundary between rhenane and linguif divisions does not appear at common times in Fig 2 for Sinsin (Belgium) and Fuhe (China). These discrepancies are easily explained if the magnetic susceptibility correlations are unreliable/misleading.

b) Correlation of magnetic susceptibility variations.

The magnetic susceptibility of sedimentary rock samples depends on many factors including the mineralogy of the bulk components, the presence of trace amounts of magnetite-like minerals and the "magnetic grain size" of these magnetite-like minerals. In some circumstances there can be a simple inverse relationship between the MS and carbonate contents provided the carbonate is non-ferroan (i.e. diamagnetic so MS nearly zero) and the main magnetisable minerals (e.g. paramagnetic clay minerals) do not include strongly magnetisable components (e.g. ferromagnetic single-domain magnetite). Correlations of MS with a basin have proved possible (e.g. for the Kimmeridge Clay Formation in Dorset, UK). In general there is no reason to suppose that the MS log in one sedimentary basin can be correlated with another basin given differences in lithology, mineralogical provenance and diagenetic histories.

Consequently, I am not convinced that the correlations indicated by red lines in Suppl. Figs 1 and 2 represent or approximate time lines since the different sections represent very different lithologies/sedimentological histories. The red lines imply correlations that would mean the biostratigraphic boundaries are not time lines. Although it is conceivable that the biostratigraphic boundaries do not provide valid time lines, there is no reason to suspect that MS logs would provide a more reliable globally applicable signal. This compromises the suggestion that the six sections have been successfully correlated and placed onto a common time scale.

The power spectra of MS in Fig 2 show very different signals. If the MS records provide global signals then why are the spectra so different? Unlike Sinsin and Fuhe, the spectra for H-32 and CG-1 provide no evidence for precession-scale variability because the peaks do not exceed the 95% confidence levels shown. [In fact the spectral backgrounds of these spectra would have been much more appropriately modelled using a power-law fit.] The spectrum for Sinsin contains no evidence for the short eccentricity cycle – again contrasting with the other sections and casting doubt on the idea that the MS signal is of global significance.

c) Correlation of $\delta^{13}\text{C}$ variations.

There is a well-known large increase in $\delta^{13}\text{C}$ near the Frasnian/Famennian boundary. This does not mean that at other times the local carbon isotope records can be correlated. By far the largest step-change in $\delta^{13}\text{C}_{\text{carb}}$ in the CG-1 (Iowa) section occurs at close to the Zone 12/13 boundary according to the data in Fig. 1. If the biostratigraphic data are reliable this implies that the CG-1 (Iowa) section involves a large increase in $\delta^{13}\text{C}$ that is not recorded by the other sections, but different $\delta^{13}\text{C}$ signals in different places invalidates the correlations shown using blue lines in Suppl. Fig. 2. Alternatively, the biostratigraphic data are unreliable at CG-1 and the F-F boundary actually occurs close to 28 m, but this also implies that the correlations indicated cannot be considered reliable. The authors need to demonstrate convincingly that small-scale variations in $\delta^{13}\text{C}$ have global significance. I do not believe that there is currently sufficient independent time control (especially from high-resolution biostratigraphy) for the correlations of $\delta^{13}\text{C}$ away from the F-F boundary to be considered good approximations to time lines.

1 Reviewer 1

Reviewer 1 Comment 1

In this contribution, the authors attempt to shed light on the pacing of paleoenvironmental events which accompanied one of the five mass extinctions at the Frasnian-Famennian (F-F) boundary. This work is firstly based on new biostratigraphy, magnetic susceptibility and carbon isotope data from four sections (or cores) across the Frasnian - Famennian boundary. These proxies are used to correlate each section to two others whose the cyclostratigraphical calibration was already published. Hence, relative depths are converted in numerical ages (based on the succession of 405 kyr eccentricity cycles), allowing the comparison of remote paleoenvironmental signals in a same temporal referential. Finally, the authors calculate the obliquity power recorded in the $\delta^{13}C$ signals of two sections and demonstrate that this orbital cycle was preponderant during the Kellwasser events responsible for the Late Devonian extinction. After several readings of the manuscript, I have a mixed feeling about this paper. On the one hand, it is well written, the new data are interesting, the Monte-Carlo intercalibration method of temporal scales is great, and the conclusions promising for our understanding of the Late Devonian events.

Response We are very pleased to read that Reviewer 1 considers our paper to represent an important advance in our understanding of the Late Devonian mass extinction event. We appreciate the constructive feedback from the reviewer, which we address point by point in this document.

Reviewer 1 Comment 2

On the other, the methodological part is confusing (with some circular reasoning), some parts presented as results are already published and bring length to the text, and I do not understand why the authors detailed and intercalibrated six sections if only two of them (whose one already published) are used to support the conclusion about the obliquity pacing of Late Devonian events.

Response To make the methodological aspects of the manuscript more transparent, we moved the correlation figures (Suppl. Figs. 1 and 2 in the previous version of the manuscript) to the main text (now Figs. 5 and 6). We also further explain the functioning of our age modeling strategy in the Supporting Information with the help of an example. More specifically, we use three distorted eccentricity-tilt-precession (ETP) series as analogues for our Devonian proxy records. We established ten tie-points (in the depth-domain) between the three distorted ETP series, and adopt the same time-calibrating approach as for the Devonian proxy series to reconstruct the original chronology.

We address the issue of circular reasoning by showing evolutive harmonic analysis (EHA) in the depth domain in Supplementary Figure 1. On this figure, we show that one can trace the imprint of eccentricity, obliquity and precession throughout the depth domain, which means that these are not *introduced* during our time-calibrating process.

Reviewer 2 has a similar comment with regards to the lengthy description of each of the six studied sections. We followed his/her suggestion to move large parts of that information to the Methods sections, so to improve readability of the Results section.

The reviewer is correct that we derive our main conclusion (i.e. the significant link between obliquity and the pacing of the Late Devonian mass extinction event) based on the carbon isotope records from the Fuhe and Kowala sections. Nevertheless, we emphasize the important role of the other four sections (Section C, H-32, CG-1 and Sinsin) in this study. All six sections are needed to come to a global astrochronology for the Late Devonian mass extinction specifically (integrating biostratigraphy, carbon isotope chemostratigraphy and magnetic susceptibility correlations), and for showing the potential for cyclostratigraphic applications in the Paleozoic, more generally. In 2013, Linda A. Hinnov wrote: "*Paleozoic cyclostratigraphy represents the next great frontier in the study of astronomical forcing*" (GSA Bulletin, v.125, no. 11/12, p. 1703-1734). Our study represents a major step in crossing this *frontier*.

Reviewer 1 Comment 3

Moreover, I think that the title is not appropriate regarding the discussion. Indeed, the authors explain that the obliquity signal was well expressed in the sedimentary series because the eccentricity power was low. As the coeval rises of $\delta^{13}C$ (and related anoxic events) are only explained by low eccentricity parameters, I conclude that the title should start with “Eccentricity pacing...”.

Response The reviewer is right in his/her comment that the original title did not reflect the important role of (low) eccentricity in determining the timing of the Late Devonian mass extinction. In response to this comment, we changed the title to “*Links between eccentricity forcing, obliquity pacing and the Late Devonian mass extinction event*”.

Reviewer 1 Comment 4

Finally, I am convinced by the data that an orbital factor likely paced the Devonian anoxic events (and organic deposits) but if I come back to the title once again, have we quantified evidences for cyclic fluctuations in diversity levels across the F-F boundary? The authors never clearly explain the orbital link with the dynamics of biodiversity crisis. I mean that orbital configurations with low eccentricity parameters are cyclic and occurred before and after the F-F boundary. So why this orbital parameter triggered lethal conditions only during these two intervals? These points are especially lacking in the text, and could bring more interest to the study, especially for a publication in Nature Communications.

Response We agree with the reviewer that -for a publication in Nature Communications- it is crucial to answer the question **why the rapid increase in eccentricity after a long-term node in the 2.4-Myr eccentricity cycle did not provoke lethal conditions every 2.4 Myr?** We answer this question in the last paragraph of the Discussion section:

During this period of low eccentricity, organic carbon on land could accumulate due to the avoidance of climates with extreme seasonality, causing marine $\delta^{13}C$ to rise at the beat of obliquity. A few tens of thousands of years later, eccentricity increased rapidly and reaches its subsequent 405-kyr maximum (Fr-LEC 17, Fa-LEC 1). This sequence of astronomical configurations supposedly contributed to a rapid warming of global climate as well as to an intensification of the hydrological cycle³⁴. Under such climatic conditions, sea level rise and increased weathering are instigated and likely contributed to the eutrophication of shallow seas, and thus to the widespread deposition of the organic-rich UKW. Yet, astronomical climate forcing is not ultimately responsible for the Late Devonian mass extinction. During the Devonian, the global carbon cycle was in a vigorous state, with enhanced silicate weathering under warm and humid conditions, and increased carbon burial through a productivity-hypoxia-nutrient recycling feedback loop. This state of vigorous carbon cycling made the Devonian quite vulnerable for ocean anoxia and mass extinction. The ultimate cause for the Late Devonian mass extinction thus probably lies with a unique convergence of processes invigorating the global carbon-cycle (e.g. evolution of land plants, volcanism, tectonic processes), but the exact timing of the outbreak of the LKW and UKW seems to be linked to a particular succession of astronomical configurations.

The reviewer rightly notices that we did not quantify cyclic fluctuations in diversity levels across the F-F boundary. The reason why we do not attempt this is twofold. First of all, biodiversity data is not available at high enough resolution to quantify a supposed obliquity imprint. Secondly, the focus of the paper lies on the paleoenvironmental changes across the F-F boundary (measured by magnetic susceptibility and carbon isotopes), their imprint of astronomical forcing, and the paleoclimate and paleoenvironmental feedback mechanisms that led to Late Devonian anoxia and organic deposits.

Reviewer 1 Comment 5

The introduction provides a good explanation of the problem. But from lines 49 to 57, I think that the authors need to tell what is really new and what is already published by the authors. A brief description (in one or two sentences) of the orbital calibration is also needed because the result parts in difficult to follow without methodological keys.

Response In the first paragraph of the Results section, we now make a clear statement on what has been previously published, and what is new. In the second paragraph of the Results section, we briefly describe how we tied-in the H-32, CG-1, Sinsin and Fuhe section into the astrochronological framework of Section C and Kowala, using magnetic susceptibility and carbon isotope correlations (respecting biostratigraphic constraints). We also moved the Figures that show these correlations to the main manuscript (now Figures 5 and 6), so that the reader gets an immediate sense of how the stratigraphic relationships between the sections have been established.

Reviewer 1 Comment 6

Line 42: The authors mention "environmental changes responsible for the mass extinction". Except anoxia and organic matter deposition, what are there?

Response Other environmental changes that have been linked with the Devonian mass extinction include fast temperatures changes, upward movement of the chemocline, sea level changes and changes in nutrient availability (e.g. Joachimski and Buggisch, 2002; Kump et al., 2005; Algeo et al. 1998, Song et al., 2017). It goes without saying that all these environmental factors are interconnected through complex cause-and-effect chains. However, we believe that at this introductory stage of the manuscript, it is better to make a general statement rather than overloading the reader with a long list of environmental changes, possibly connected to the mass extinction.

Reviewer 1 Comment 7

I acknowledge the authors in their effort to propose a global temporal calibration of sections. However, some points may be further discussed or explained:

First, basic information regarding sampling, number of new data, and outcrops are completely lacking. This is one important aspect of the work, which deserves to be mentioned. Moreover, I suggest moving facies and paleoenvironmental descriptions of cores and sections in the Methods, as the sedimentology of most of them is already described in publications.

Response As suggested by Reviewers 1 and 2, we moved facies and paleoenvironmental descriptions of all sections to the Methods sections. There, we also added some of the basic information that is requested by the reviewer: the geographic coordinates of the different sections, the sampling interval for the different proxy series, the total number of data points for each proxy series (both for new and previously published records).

Reviewer 1 Comment 8

The authors correlate the outcrops and cores by using magnetic susceptibility data and $\delta^{13}C$ data from remote domains. Correlation lines are shown on supplementary figures but I wonder how some correlation lines are drawn. For me, magnetic susceptibility signals are rarely global and may present local variations or peaks (highly dependent on sampling resolution and facies). For example, the authors mention that debris flows are present in the Fuhe section. . . How to deal with? Moreover, the correlation of MS peaks is sometimes very subjective on figures. . . I suggest to the authors: 1) to add biostratigraphical scheme of each section to have a better appraisal of correlations; 2) to better explain how MS and $\delta^{13}C$ peaks are correlated; 3) to further explain how the Monte Carlo approach can help to measure errors in the position of tie-points; 4) to discuss how difference in the sampling effort of section can affect correlations.

Response In response to this comment, we followed the suggestion of the reviewer and added the biostratigraphical scheme of each section to Figures 5 and 6 (Supplementary Figures in the previous version of the manuscript). Also note that we added the biostratigraphic scheme for section C to Figures 2 and 3 in the time domain. We also changed the color coding of the correlating lines in Figures 5 and 6. We now use colored lines (red or blue) when the correlating tie was drawn based on the data shown in the Figure (magnetic susceptibility or carbon isotopes). The other correlating ties are shown in grey to be less outstanding. In the revised version of the manuscript, we now discuss which correlating ties do not respect biostratigraphic constraints. For the two instances where the latter is the case, we provide some arguments why we prefer our proposed correlations (see also Figure 3 in this document, in response to Reviewer 3 and Supplementary Figure 5). With these changes, we aim to be completely transparent about our reasons *why* we drew certain correlation lines. Each of the correlating lines has been well-considered and different options have been looked at by the authors.

The Supplementary Information, in which the Monte Carlo approach for time modeling is illustrated with ETP-analogues for the Devonian proxy records, provides the interested reader with a look under the hood of our Monte Carlo approach. The ETP illustration is built around

three comprehensive figures. This means that one does not necessarily need to have experience with programming in R to understand how our Monte Carlo approach works. However, for those who possess that experience, we make our R scripts available through GitHub.

Reviewer 1 Comment 9

I understand that the authors apply the well-defined orbital calibration of the Canadian and Polish sections to the other cores to define a common temporal scheme. However, I was surprised that they never analyzed the raw MS patterns of new studied sections to show that the 405 kyr cycles used for relative dating are well recorded. Instead, they analyzed the MTM power spectra of time-corrected MS records to show that expected eccentricity, obliquity, and precession frequencies are well present. Is it not a circular reasoning? We could suggest that a previous time calibration (by 405 Myr cycles) artificially introduce expected high-frequency peaks, isn't it? The authors have to better explain these points in the method and show the spectral analyses of raw MS data.

Response Figure S1 shows the evolutive harmonic analyses (EHA) of H-32, CG-1, Sinsin and Fuhe magnetic susceptibility series in the depth domain. The different astronomical components can be traced in the depth domain (white dashed lines on that figure), demonstrating that these have not been introduced by the age modeling strategy adopted in this paper. Therewith, we significantly attenuated the risk of circular reasoning in our age-modelling approach. We acknowledge the risk for circular reasoning in the manuscript, refer the reader to the supplementary figure that shows that the astronomical imprint in the susceptibility signal is original, and thus not introduced by the age modeling strategy.

Reviewer 1 Comment 10

On Figure 2, I remark that, contrarily to what explained in the text, the expected eccentricity, obliquity, and precession cycles are sometimes below the 95% confidence interval (in Sinsin for example). This is important to mention in the text and propose explanation (local paleoenvironmental influence?).

Response The magnetic susceptibility time series in Figure 2 were re-calibrated, in comparison to the previous version of the manuscript. Indeed, some correlation lines were redrawn so to be in better agreement with the biostratigraphy (see Reviewer 3, Comment 1). Therefore, the MTM power spectra in Figure 2 have also been recalculated. We made sure that the text and the Figure are in agreement, and we discuss the power spectra in somewhat more detail. For example, we discuss the mismatch between the obliquity-related peak in the Fuhe section (at 0.036 cycles/kyr) and the expected frequency of Devonian obliquity (0.031 cycles/kyr).

Reviewer 1 Comment 11

The age modelling method described at lines 253-265 is very original. However, I think that a supplementary figure with examples would help to clarity.

Response As mentioned before, we illustrate the age modelling method in the Supplemen-

tary Information using ETP series. This analogue for the Devonian proxy series provides the interested reader with a look under the hood of our Monte Carlo approach. The ETP illustration is built around three comprehensive figures. This means that one does not necessarily need to have experience with programming in R to understand how our Monte Carlo approach works. However, for those who possess that experience, we make our R scripts available through GitHub.

Reviewer 1 Comment 12

In my opinion, the description of cores and sections is very long and sometimes boring since some results (of Polish and Canadian sections) are already published. I suggest shortening this part by further focusing on the new data (including biostratigraphy, MS data, and $\delta^{13}\text{C}$). Other points may be included in the Method.

Response To improve the readability of the main manuscript, we moved facies and paleoenvironmental descriptions to the Methods sections. There, the reader can also find information on sampling resolution and number of datapoints for magnetic susceptibility, carbon isotope measurements, as well as for biostratigraphy. In the Figure caption of Figure 1, we provide a short overview of which data was produced in the framework of this paper (MS, $\delta^{13}\text{C}$ and biostratigraphy), and which data was previously published (with the appropriate references).

Reviewer 1 Comment 13

L74: I do not think that the potassium content is useful as the authors never describe it or use it as correlation tool. Maybe to remove in the figure 1.

Response We removed the potassium content data-series of the Kowala section from Figure 1.

Reviewer 1 Comment 14

L159-162: The link between eccentricity, sea level, and warming event is unclear. Please, better explain with bibliographical references showing this link.

Response In the revised version of the manuscript, we cite Bond and Wignall (2008), who reviewed (and then supported) the link between transgression-anoxia-extinction, originally proposed by Johnson et al. (1985). The link between warm and humid climates and high eccentricity is supported by the climate modeling cited (De Vleeschouwer et al., 2014). This phase-relationship is the same as the phase-relationship between global climate and eccentricity observed in the Cenozoic (De Vleeschouwer et al., 2017).

Reviewer 1 Comment 15

L166: Is this hypothesis coherent with the recent papers of Eldrett et al. (2015) “Origin of limestone-marlstone cycles: Astronomic forcing of organic-rich sedimentary rocks from the Cenomanian to early Coniacian of the Cretaceous Western Interior Seaway, USA, EPSL

Response Yes, the model put forward in Eldrett et al. (2015, Figs. 9 and 10 therein) is very similar to the model presented by Martinez and Dera (2015, Fig. 4 therein). Hence, we now cite Eldrett et al. (2015) together with Martinez and Dera (2015), Batenburg et al. (2016), Sprovieri et al. (2013) and Laurin et al. (2015) in the following sentence: *Several recent studies pointed to the link between long-term eccentricity forcing on the one hand and the Myr-scale behaviour of the Mesozoic carbon cycle on the other hand.*

Reviewer 1 Comment 16

L183: It is disturbing to analyze the obliquity power along two sections to conclude that this is related to a low eccentricity... Is it possible to measure the eccentricity power for the 6 locations by focusing on eccentricity (<0.02) or precession frequencies? A test on the H-32 Iowa core would be maybe more relevant to show the patterns on the long term.

Response The results of the obliquity power analysis in Figure 4 actually strongly suggests the link with eccentricity. Indeed, the obliquity power of the Fuhe section exhibits two strong increases: A first time at the eccentricity minimum that separates Fr-LEC 15 and Fr-LEC 16, and -more importantly- a second time at the eccentricity minimum separating Fr-LEC 16 and 17. The latter increase in obliquity power is associated with the onset of the UKW. This argument is now included in the revised version of the manuscript. We also added the blue/white horizontal bands that designate the Fr-LEC cycles to Figure 4 in order to make the obliquity-eccentricity link more clear to the reader.

2 Reviewer 2

Reviewer 2 Comment 1

Overall this is an important manuscript that deserves publication. The manuscript could benefit from some improvements in the presentation of language, particularly in the front half of the manuscript. Overall I have mostly linguistic suggestions, however, I think the authors can substantially improve the front of the manuscript. In particular, it currently is disjointed in the beginning. It almost reads more as a list of facts than as a coherent manuscript. The back half of the manuscript, once you get to the discussion reads much more comfortably. I would suggest either including much of the details about each specific section in the supplementary materials section, or figure out some other way to make this less like a Gregorian chant of information. The interesting bits are presented all at once in the end of the manuscript with little to no warning that there is going to be that much of interest. In short, there is nothing up front that begs the reader to keep reading. The authors can fix this quite easily I would imagine and presage where you are going with the front of the manuscript. Move much of the other information, which is too short to be terribly useful anyhow, to the supplementary sections. When you do that, it will also allow you to increase the amount of detail you can include about each section. I strongly believe this manuscript should be published following these minor revisions. Below is a list of specific comments on the manuscript by line number.

Ln. 23: Throughout the manuscript, please be careful with chronostratigraphic vs geochronologic terms. Because you are talking about stratigraphy here, you need to use Upper Devonian. Similarly, please refrain from using the word ‘period’ colloquially. This word has a specific geologic meaning. I would suggest that you replace it with the word ‘interval’.

Ln 27: Again, don’t use ‘periods’ here.

Response The paper was reorganized according to the suggestions of Reviewers 1 and 2. The description of the different sections and cores has been moved to the Methods section, the Introduction has been rewritten significantly, and the Results and Discussion sections have been expanded.

Both instances of "periods" were replaced by "intervals"

Reviewer 2 Comment 2

Ln. 24: I would suggest you use 'orbitally-tuned' or 'orbitally-calibrated'
 Ln 26: Unless instructed by the journal, it is not normal to put the $\delta^{13}\text{C}$ in italics. Please make it regular text. Please fix this throughout the manuscript.
 Ln. 40: Since you are talking about time here, please use Middle to Late Devonian.
 Ln 60: We studied. . . this is past tense.
 Ln. 64: replace 'passage' with 'interval'
 Ln. 66: the word 'episodic' is problematic. Please remove.
 Ln. 67: . . . content, but the two black shale. . .
 Ln. 96: A section cannot possess. . . Please rewrite this sentence.
 Ln. 103: Again, do not anthropomorphize stratigraphic section. . . a core cannot hold anything. Please rewrite.
 Ln. 133: . . . ends halfway through Fr-LEC14
 Ln 176: . . . few tens of thousands. . . Same again on line 197

Response All these suggestions were adopted in the revised version of the manuscript.

Reviewer 2 Comment 3

Ln. 44: Please correct this to 'constraints on the calibration of the stage boundaries within the Devonian System. . .'
 Ln. 44: Please remove 'therewith'. . . it is awkward.

Response This sentence was rephrased to "*Recently, however, cyclostratigraphic efforts yielded constraints on the numerical ages of stage boundaries within the Devonian System*11-14. *These studies provided a new chronometer for this part of the Paleozoic, based on the most stable astronomical cycle, i.e. the 405-kyr long eccentricity cycle.* "

Reviewer 2 Comment 4

Ln. 50: duration of an entire age. . .

Response The sentence in question does no longer occur in the revised version of the manuscript

Reviewer 2 Comment 5

Ln. 112: Please remove 'which makes that'. . . Also, this is an example of the type of detail that belongs in the supplementary section. This reference to meterage here comes out of nowhere and can be much better dealt with more thoroughly in the supplementary materials section.

Response All the section-specific information was moved to the Methods section of the manuscript, as suggested by Reviewers 1 and 2. We also rephrased this sentence according to the suggestion of Reviewer 2.

Reviewer 2 Comment 6

Ln. 137: please do not abbreviate here... what is resp.?

Response We spelled out ‘respectively’ and placed it at the end of the clause.

Reviewer 2 Comment 7

Ln. 152: Replace ‘height’... to what are you referring? Stratigraphic height? Maximum isotope values? Stratigraphic thickness?

Response We were referring to maximum isotope values. To avoid ambiguity, we replaced “height” by “peak”.

Reviewer 2 Comment 8

Figures: I would prefer it if you included headings for each of your columns... Stage, Conodont Zone, etc... also, you need to include a legend for your stratigraphic columns.

Response We did not include headings to Figure 1, as we judged that it is sufficiently clear that the first column represents Stages and the second column conodont zones. However, we did use the same lithological column style for all sections in the revised version of Figure 1. We also included a legend for our stratigraphic columns in Figure 1.

3 Reviewer 3: Graham P. Weedon

Reviewer 3 Comment 1

(a) Biostratigraphic constraints not followed.

On lines 120 to 123 it is stated: “We tie the four sections (H-32, CG-1, Sinsin and Fuhe) into the common astrochronological framework of Section C and Kowala by correlating distinct features in magnetic susceptibility (red ties on Suppl. Fig. 1) and carbon isotope geochemistry (blue ties on Suppl. Fig. 2).” This approach makes sense because biostratigraphic boundaries are much more likely to approximate time lines than magnetic susceptibility or $\delta^{13}C$ variations.

However, comparing Fig. 1 with Fig. 2 it is clear that the authors have not stuck to their procedure. For example, the Zone 11/12 boundary of H-32 (Iowa) is located at Fr-LEC 11 in Fig. 2 whereas the same boundary at CG-1 (Iowa) occurs in Fr-LEC 14. Similarly the Zone 12/13 boundary of H-32 occurs in Fr-LEC 13 in Fig 2, but in Fr-LEC 15 for CG-1 and at a different time in Fr-LEC 15 for Section C (western Canada). Also the boundary between rhenane and linguif divisions does not appear at common times in Fig 2 for Sinsin (Belgium) and Fuhe (China). These discrepancies are easily explained if the magnetic susceptibility correlations are unreliable/misleading.

Response Graham P. Weedon is correct in his observation that not all of the correlating lines are in perfect agreement with the biostratigraphic constraints.

First of all, in order to allow the reader to evaluate the discrepancies between biostratigraphy and correlation lines based on magnetic susceptibility or $\delta^{13}C$, we added the biostratigraphical scheme of each section to Figures 5 and 6 (previously Suppl. Figures 1 and 2). In response to Graham P. Weedon's comment, we revisited the biostratigraphic data for all sections: this resulted in three small adjustments in comparison with the previous version of the manuscript.

- My co-author James E. Day did a personal restudy of all slides from H-32 (original biostratigraphy was done by a student) and found that the Zone 12 - Zone 13 boundary needed to be moved down by 1.5 meter. This implies that the Zone 11 and 12 interval in the Sweetland Creek Shale in the H-32 core (as well at the type section of the Sweetland Creek Shale) is highly condensed. Most of the Sweetland Creek Shale section represents and expanded record of the post LKE interval through the very Early Famennian.
- At Section C, the Zone 11 - 12 boundary has never been observed, because the lowermost conodont sample in Section C occurs at 320.5 m (assigning this level to Zone 12). This biostratigraphic dataset can be found in Table 4 in Whalen and Day (2008; note the 320 m offset between the meter scale in their Table 4 on the one hand, and the MS curve in their Figure 9 and our Figure 1 on the other hand). This means that the lower 40 meters of Section C are unconstrained by biostratigraphy. But, we can use the cross-basin correlations by Whalen and Day (2010) to tentatively place the Zone 11 - Zone 12 boundary. Whalen and Day (2010) documented 24 Frasnian and 1 earliest Famennian magnetic susceptibility "events" and demonstrated high resolution, graphic correlation of the susceptibility signature across the western Canada sedimentary basin. According to Figure 9 in Whalen and Day (2010), the Zone 11 - 12 boundary occurs in "event" F21. F21 in Sections MC and KC corresponds to the Fr-LEC 12/13 boundary in De Vleeschouwer et al. (2012). See Figure 1 on the next page of this document for illustration. Following that reasoning, we can suggest the Zone 11-12 boundary to be expected at 317 m in Section C. Hence, in the revised version of the manuscript, we suggest the Zone 11 - 12 boundary at Section C (with a dashed line) at the Fr-LEC 12-13 boundary in the time domain, and at 317 m in the depth domain.
- The biostratigraphic zonation for the Sinsin section comes from Sandberg et al. (1988). In this original work, Charles Sandberg never identified the *rhenana* - *linguiformis* boundary (see Figure 2 in this document). However, when Kaiho et al. (2013) published on the Sinsin section, these authors drew a tentative lower *linguiformis* boundary 1.5 m below the F-F boundary. In the previous version of the manuscript, we mistakenly took over their tentative boundary as a clear *rhenana* - *linguiformis* boundary. In the present version of the manuscript, we indicate that the lower part of the Sinsin section is unconstrained by biostratigraphy, by drawing a dashed line 1.5 m below the F-F boundary. Moreover, we no longer mention the *rhenana* conodont Zone for the lower part of the Sinsin section, but instead put a question mark in this part of the biostratigraphical column.

Secondly, we redrew some correlating lines in Figures 5 and 6 (previously Suppl. Figures 1 and 2). This was necessary mainly because of the revision of the H-32 biostratigraphy described above. As specified in the manuscript, these correlating lines were drawn, respecting the biostratigraphic constraints. However, in two instances there was no other option than to deviate -to a limited extent- from the biostratigraphic zonation scheme. Specifically, we

FIG. 11.—Cross section with spline-smoothed δMS data from each measured section, Late Devonian third-order sea-level events of Johnson et al. (1985, 1996), as modified by Day et al. (1996), Day (1998), and Whalen and Day (2008), depositional sequences (Whalen and Day 2008), and sea-surface temperatures determined from oxygen isotopes from conodont apatite (Jochimski et al. 2009). Note that most sequence boundary intervals (upper HST, LST, lower TST) are characterized by MS highs, while upper TSTs and lower HSTs record MS lows. Also note the major increase in MS values associated with rising temperatures during the Frasnian.

Figure 1: By combining the correlation of biostratigraphic zonation scheme with MS events from Whalen and Day (2010; Fig. 11), and the cyclostratigraphic correlation by De Vleeschouwer et al. (2012), the Zone 11 - 12 boundary can be tentatively placed at 317 m in Section C.

FIGURE 10. - Detailed columnar section of thin Matagne Shale equivalent (bed 12 and overlying carbonaceous shale) and adjacent parts of underlying Neuville Formation (Frasnian) and overlying Famenne Shale (Famenian) at Sinsin roadcut, Belgium. Key beds are from COEN (1973). Sequential event-stratigraphic interpretation of important samples and lithologic changes related to late Frasnian mass extinction are given to left of column. Numbers to right of conodont sample numbers are percentages of three genera in heading. Normal late Frasnian conodont faunas before extinction event in this outer-shelf paleotectonic setting are assigned to deep-water palmatolepid-polygnathid biofacies.

Figure 2: Original biostratigraphic analysis of the Sinsin section from Sandberg et al. (1988)

correlate the stratigraphic levels at 19 and 21.5 m at the CG-1 core with 320 and 330 m at Section C. These levels occur in Frasnian Conodont Zone 11 in the CG-1 core, but occur in Frasnian Conodont Zone 12 in Section C. In this case, it is unclear whether this discrepancy is caused by diachronism between Iowa and western Canada, an inaccurate biostratigraphic zonation, or erroneous correlation. Yet, the excellent match between the susceptibility signals of these two signals in this interval (Suppl. Fig. 2) strongly substantiates our preferred correlating lines (Fig. 5). The second instance consists of the correlation between the stratigraphic levels at 18 and 21.5 m in Fuhe (in the *linguiformis* Zone) with the stratigraphic levels at 355.5 and 359 m at Section C (in the *rhenana* Zone). Given the good chemostratigraphic control in the latter instance, it is most likely that the *rhenana* - *linguiformis* boundary is identified too low in the Fuhe section. All these features are discussed in the revised version of the manuscript.

Reviewer 3 Comment 2

(b) Correlation of magnetic susceptibility variations.

The magnetic susceptibility of sedimentary rock samples depends on many factors including the mineralogy of the bulk components, the presence of trace amounts of magnetite-like minerals and the “magnetic grain size” of these magnetite-like minerals. In some circumstances there can be a simple inverse relationship between the MS and carbonate contents provided the carbonate is non-ferroan (i.e. diamagnetic so MS nearly zero) and the main magnetisable minerals (e.g. paramagnetic clay minerals) do not include strongly magnetisable components (e.g. ferromagnetic single-domain magnetite). Correlations of MS with a basin have proved possible (e.g. for the Kimmeridge Clay Formation in Dorset, UK). In general there is no reason to suppose that the MS log in one sedimentary basin can be correlated with another basin given differences in lithology, mineralogical provenance and diagenetic histories.

Consequently, I am not convinced that the correlations indicated by red lines in Suppl. Figs 1 and 2 represent or approximate time lines since the different sections represent very different lithologies/sedimentological histories. The red lines imply correlations that would mean the biostratigraphic boundaries are not time lines. Although it is conceivable that the biostratigraphic boundaries do not provide valid time lines, there is no reason to suspect that MS logs would provide a more reliable globally applicable signal. This compromises the suggestion that the six sections have been successfully correlated and placed onto a common time scale.

The power spectra of MS in Fig 2 show very different signals. If the MS records provide global signals then why are the spectra so different? Unlike Sinsin and Fuhe, the spectra for H-32 and CG-1 provide no evidence for precession-scale variability because the peaks do not exceed the 95% confidence levels shown. [In fact the spectral backgrounds of these spectra would have been much more appropriately modelled using a power-law fit.] The spectrum for Sinsin contains no evidence for the short eccentricity cycle – again contrasting with the other sections and casting doubt on the idea that the MS signal is of global significance.

Response

We agree with Graham P. Weedon that the correlation of magnetic susceptibility variations

Figure 3: Comparison of magnetic susceptibility signals in the time domain, in response to the doubts raised by Graham P. Weedon concerning the use of magnetic susceptibility as a correlation tool. Part of this Figure is included as Supplementary Figure 2.

does not offer the same unambiguous timelines as the chemostratigraphic correlation of $\delta^{13}C$ excursions. However, in the present Late Devonian case, there is reason to suppose that the MS log in one sedimentary basin can be correlated to another basin. There are quite a few papers published on this topic, especially in the Devonian, demonstrating the global correlative potential of magnetic susceptibility variations. In the revised version of the manuscript, we acknowledge the discussion that exists on whether/when one can use magnetic susceptibility as a correlative tool, and point the reader to relevant previous work in the Devonian with the following sentences:

The determination of stratigraphic relationships between globally-distributed sections based on carbon isotope variations is well established²¹, but interbasinal correlations based on magnetic susceptibility data are not undisputed. This is because the magnetic susceptibility of a rock sample depends on the concentration and type of magnetic minerals (often assumed to be a function of sea-level and climate), as well as the size and shape of the magnetic grains. Nevertheless, several Devonian examples exist of valid global correlations based on magnetic susceptibility, despite differences in palaeogeographic setting, facies, accumulation rate or depositional history (Crick et al., 1997; Ellwood et al., 2000; Boulvain et al., 2010; Whalen et al., 2015), suggesting a forcing mechanism operating at the global scale.

Moreover, the data presented within this paper, clearly shows that magnetic susceptibility variations present the opportunity to establish timelines between widely separated sections. As an example, we show a direct overlay of the magnetic susceptibility signals of the Sinsin section and the Fuhe section in the time domain, as well as a direct overlay of the susceptibility signals of CG-1 and Section C. There are unmistakably far-reaching parallels between the temporal evolution of these time series (Figure 3 in this document, Supplementary Figure 2), which we consider a strong enough argument to continue using magnetic susceptibility variations as a correlating tool. For that reason, we write in the revised version of the manuscript: *For example, Section C, H-32, Sinsin and Fuhe exhibit a characteristic double minimum in magnetic susceptibility just prior to the F-F boundary (Fig. 1). This susceptibility signature corroborates the utility of magnetic susceptibility as a correlative tool in the framework of this Late Devonian study.*

Reviewer 3 Comment 3

(c) Correlation of $\delta^{13}C$ variations.

There is a well-known large increase in $\delta^{13}C$ near the Frasnian/Famennian boundary. This does not mean that at other times the local carbon isotope records can be correlated. By far the largest step-change in $\delta^{13}C_{carb}$ in the CG-1 (Iowa) section occurs at close to the Zone 12/13 boundary according to the data in Fig. 1. If the biostratigraphic data are reliable this implies that the CG-1 (Iowa) section involves a large increase in $\delta^{13}C$ that is not recorded by the other sections, but different ^{13}C signals in different places invalidates the correlations shown using blue lines in Suppl. Fig. 2. Alternatively, the biostratigraphic data are unreliable at CG-1 and the F-F boundary actually occurs close to 28 m, but this also implies that the correlations indicated cannot be considered reliable. The authors need to demonstrate convincingly that small-scale variations in $\delta^{13}C$ have global significance. I do not believe that there is currently sufficient independent time control (especially from high-resolution biostratigraphy) for the correlations of $\delta^{13}C$ away from the F-F boundary to be considered good approximations to time lines.

Response In this paper, we propose 5 chemostratigraphic correlations based on the carbon isotope stratigraphy of the studied sections. All of these correlations are limited to the interval of carbon cycle perturbation, from the onset of the LKW till the stabilisation of the global carbon cycle during the aftermath of the UKW. Indeed, the oldest chemostratigraphic timeline is assigned a relative age 774 kyr older than the F-F boundary and the youngest chemostratigraphic timeline is assigned an age 488 kyr younger than the F-F boundary. In other words, the extent of our correlations of carbon isotope variations is limited to a 1.26 Myr interval, spanning both Kellwasser events and spanning the F-F boundary. The large step-change in $\delta^{13}C_{carb}$ in CG-1 is interpreted as the expression of the LKW in a shelf-margin paleoenvironmental setting of the Iowa Basin. This interpretation is supported by the work of Sliwinski et al. (2011), who demonstrated a larger positive carbon isotope excursion on the platform top than on the slope or basin for the *punctata* event. Hence, we correlate this excursion to its time-equivalent $\delta^{13}C$ expressions of the LKW in the deeper-water settings of the other studied sections.

Sliwinski, M.G., Whalen, M.T., and Day, J.E., 2011, Stable Isotope and Trace Element Anomalies during the Late Devonian ‘punctata Event’ in the Western Canada Sedimentary Basin, *Palaeogeography, Palaeoclimatology, Palaeoecology*, v. 307, p. 245-271.

REVIEWERS' COMMENTS:

Reviewer #1 (Remarks to the Author):

This is my second review of the manuscript "Links between eccentricity forcing, obliquity pacing and the Late Devonian mass extinction event" by De Vleeshouwer et al. Overall, I am glad to see that the authors followed all the recommendations of reviewers. Important changes have been done : appraisal of figures, better descriptions of methods, and better discussion of links between orbital and biotic process. The text is now very clear and I think that the new manuscript is now ready to be published in Nature Communications.

Reviewer #3 (Remarks to the Author):

The revised manuscript is now far easier to follow and the movement of much detail out of the main text is beneficial. The authors have acknowledged the issue raised concerning correlation of MS crossing biostratigraphic constraints. After reviewing the biostratigraphic constraints available they have provided an appropriate treatment of the residual issues in the main text. Similarly, they have directly tackled the controversial use of MS for extra-basinal/global correlation (though I would like a credible explanation). The precise use of $\delta^{13}\text{C}$ for correlation has also been clarified as restricted to a short interval centred on the F-F boundary. This means we now have a clear understanding of the correlations made and the reasoning involved so that others can test their conclusions in the future.

The power spectral evidence for regular cyclicity has been revisited so that now there is modest evidence (spectral passing the AR1 model 95% CI) for sub-405 kyr orbital forcing at the various sites. The addition of wavelet spectra for the depth domain MS (Fig S1) is useful in terms of documentation, though I'm not convinced that the the authors can justify the positions of the white dashed lines denoting different orbital frequencies in all cases and all stratigraphic levels.

I support the authors' suggestion that Figs 5 and 6 be moved to 2 and 3. Alternatively Fig 5 could replace Fig 1 if the palaeogeography map is included and the lithostratigraphic data added to the MS profiles. If Fig 1 is retained then the plot of the Polish data should be placed next to the Belgian data while the Chinese data should be moved up. In Fig 2 it would be clearer for the reader if all the Fa-LEC# and Fr-LEC# labels were provided in black.

graham.weedon@metoffice.gov.uk 13th Oct 2017